# Density Functional Theory Calculations and Molecular Docking Analyses of Flavonoids for Their Possible Application against the Acetylcholinesterase and Triose-Phosphate Isomerase Proteins of *Rhipicephalus microplus*

**DOI:** 10.3390/molecules28083606

**Published:** 2023-04-20

**Authors:** Nosheen Malak, Bader S. Alotaibi, Afshan Khan, Adil Khan, Shakir Ullah, Nasreen Nasreen, Sadaf Niaz, Chien-Chin Chen

**Affiliations:** 1Department of Zoology, Abdul Wali Khan University Mardan, Mardan 23200, Pakistan; 2Department of Laboratories Sciences, College of Applied Medical Sciences, Shaqra University, Alquwayiyah 15273, Saudi Arabia; 3Department of Botany and Zoology, Bacha Khan University, Charsadda 24420, Pakistan; 4Department of Biotechnology and Bioindustry Sciences, College of Bioscience and Biotechnology, National Cheng Kung University, Tainan 701, Taiwan; 5Department of Pathology, Ditmanson Medical Foundation Chia-Yi Christian Hospital, Chiayi 600, Taiwan; 6Department of Cosmetic Science, Chia Nan University of Pharmacy and Science, Tainan 717, Taiwan; 7Ph.D. Program in Translational Medicine, Rong Hsing Research Center for Translational Medicine, National Chung Hsing University, Taichung 40227, Taiwan

**Keywords:** density functional theory, flavonoids, molecular docking, *Rhipicephalus microplus* acetylcholinesterase 1 (RmAChE1), *Rhipicephalus microplus* triose-phosphate isomerase (RmTIM), tick, tick-borne disease

## Abstract

Ticks and tick-borne diseases constitute a substantial hazard to the livestock industry. The rising costs and lack of availability of synthetic chemical acaricides for farmers with limited resources, tick resistance to current acaricides, and residual issues in meat and milk consumed by humans further aggravate the situation. Developing innovative, eco-friendly tick management techniques, such as natural products and commodities, is vital. Similarly, searching for effective and feasible treatments for tick-borne diseases is essential. Flavonoids are a class of natural chemicals with multiple bioactivities, including the inhibition of enzymes. We selected eighty flavonoids having enzyme inhibitory, insecticide, and pesticide properties. Flavonoids’ inhibitory effects on the acetylcholinesterase (AChE1) and triose-phosphate isomerase (TIM) proteins of *Rhipicephalus microplus* were examined utilizing a molecular docking approach. Our research demonstrated that flavonoids interact with the active areas of proteins. Seven flavonoids (methylenebisphloridzin, thearubigin, fortunellin, quercetagetin-7-*O*-(6-*O*-caffeoyl-β-d-glucopyranoside), quercetagetin-7-*O*-(6-*O*-*p*-coumaroyl-β-glucopyranoside), rutin, and kaempferol 3-neohesperidoside) were the most potent AChE1 inhibitors, while the other three flavonoids (quercetagetin-7-*O*-(6-*O*-caffeoyl-β-d-glucopyranoside), isorhamnetin, and liquiritin) were the potent inhibitors of TIM. These computationally-driven discoveries are beneficial and can be utilized in assessing drug bioavailability in both in vitro and in vivo settings. This knowledge can create new strategies for managing ticks and tick-borne diseases.

## 1. Introduction

Ticks are hematophagous vertebrate ectoparasites that transmit bacterial, viral, and protozoan infections [1]. Cattle farmers worldwide are becoming increasingly concerned about the spread of the southern cow tick, *Rhipicephalus microplus* (formerly *Boophilus microplus*). When a cow tick bites an animal, it can transmit disease-causing microbes such as *Babesia bovis*, *Babesia bigemin*, and *Anaplasma marginale*, causing the animal to grow slowly, produce less milk, or even die [2]. *R. microplus* control has emerged as a central concern and area of attention in animal parasitology research and technology development due to its global distribution and impact on cattle husbandry [3]. While chemical acaricides have effectively reduced *R. microplus* populations, their widespread use has resulted in acquired resistance, making population control less likely [4]. Multiple cases of *R. microplus* acaricide resistance have emerged in recent years, making the global spread of acaricide resistance a major concern for the cattle industry [5]. As a result, research into new chemicals with distinct mechanisms of action and acaricidal efficacy against *R. microplus* is ongoing.

Moraes et al. investigated the physiological mechanisms underlying tick embryo formation to develop new tick population control strategies [6]. They found that morphogenetic alterations in *R. microplus* embryos were linked to the predominant energy source [6]. While *R. microplus* embryos develop in 21 days, they are syncytia until the fifth day. On the sixth day, zygotic expression becomes active, and the embryo assumes control over its development [7]. There seem to be two separate metabolic stages during cellularization, with major changes occurring in the glycolytic and gluconeogenic pathways [8].

Triose-phosphate isomerase (TIM), an enzyme involved in glycolysis and gluconeogenesis, catalyzes the interconversion of glyceraldehyde-3-phosphate and dihydroxyacetone phosphate. The therapeutic development of TIM has shown promise against several human-pathogenic parasites [9,10]. These include *Plasmodium falciparum*, *Trypanosoma brucei*, *Trypanosoma cruzi*, *Giardia lamblia*, and *Fasciola hepatica*. TIM is a (β/α)_8_-barrel protein, a core of eight parallel β-strands and eight α-helices on all sides. Banner et al. found TIM’s “TIM-barrel” crystal structure [11]. Moraes et al. were the first to investigate TIM’s kinetic and structural properties from embryos of *R. microplus* [6]. Their study suggested that many cysteine residues in *R. microplus* TIM (RmTIM) could be used to make antiparasitic drugs. In addition, selective inhibitors can be created by targeting this enzyme’s less conserved interface [12].

Acetylcholinesterase (AChE1) (E.C. 3.1.1.7.) is a synaptic enzyme and a major target of both organophosphates (OPs) and carbamates [13]. It is essential for the transmission of nerve impulses in both vertebrates and arthropods [14,15]. AChE1 inhibits nerve impulses by breaking acetylcholine into acetate and choline [16]. AChE1 is inhibited by the acaricides carbamate and OPs. Carbamates and OPs pesticides both inhibit AChE1 activity by binding to the esterasic and anionic sites of the enzyme. Both suppress the enzyme’s activity in a similar mode of action [17,18]. Various flavonoids and their derivatives have been found to possess inhibitory potential on AChE in *Caenorhabditis elegans* and *Spodoptera litura* [19], as well as electric eels (*Electrophorus electricus*) [20].

The majority of the organic compounds produced by plants do not appear to contribute to their growth and development. The distribution of secondary metabolites, or flavonoids, varies considerably between taxonomic plant groups [21]. The interaction of flavonoids with proteins and nucleic acids has resulted in the development of numerous bioactive compounds with pharmacological, antibacterial, and insecticidal properties. Since flavonoids are used as pesticides in medicine and agriculture, they are significant. Therefore, they could be useful in a pest control program [22].

The primary objective of computational approaches such as molecular docking and in silico absorption, distribution, metabolism, excretion, and toxicity (ADMET) analyses is to screen potential flavonoids against RmTIM and *R. microplus* AChE1 (RmAChE1) in numerous databases and libraries. Computational screening reduces the time and resources required for experimental testing in pharmaceutical research. Utilizing computational designs and molecular information on flavonoid compounds, research into the development of effective alternatives to currently used acaricides may benefit from the wide variety of flavonoids by selecting the most promising compounds. Additionally, medicinal chemistry is frequently used to identify new bioactive molecules with acaricidal applications. For example, to identify RmTIM and RmAChE1 inhibitors, natural substances from numerous databases and libraries were screened using knowledge of parasiticides and enzyme inhibitors.

## 2. Results

### 2.1. The Structural Models of TIM and RmAChE1

TIM crystallographic enzyme structures have been obtained from the Protein Data Bank (PDB) repository (PDB code: 3TH6). Using the Python molecular viewer tool of PyMOL (v1.9, http://www.pymol.org (accessed on 10 June 2022)), water molecules were removed from the protein structure, and non-polar hydrogen molecules were added. The amino acid sequence of RmAChE1, with accession number A0A0F6P2D6, was retrieved from the UniProt database (found at http://www.uniprot.org (accessed on 10 June 2022)), which is a comprehensive source for protein sequences. The Basic Local Alignment Search Tool (BLAST) analysis revealed the existence of 100 protein sequences that showed significant similarities. Out of these, 48 were AChE1s from other species, 21 were butyrylcholinesterases, and the remaining 31 were proteins that shared some degree of identity with RmAChE1. The crystal structure of mouse AChE1 (PDB: 5DTI, Chain A) was selected as the template among the 48 AChEs due to its highest resolution of 2 Å. Additionally, this structure showed a high level of similarity to RmAChE1, with 44% identity and 89% query coverage. Figure 1 demonstrates the three-dimensional (3D) structures of RmTIM and RmAChE1.

### 2.2. Validation of the Modeled Structures

To optimize 3D models for structural inspections, PROCHECK created a Ramachandran map. Non-glycine and non-proline residues’ contributions to the Ramachandran map’s u and w distribution were summarized in Figure 2A. The protein structural models were checked against protein structures in the PDB using the ProSA-web server based on z-scores (Figure 2C) [23]. The application provided the Z-score of the input structure and a plot of the residue energies. SWISS-MODEL’s Z-score for representing AChE1 showed a −9.44, indicating excellent overall model quality. Figure 2B displays the ERRAT-verified 3D structures. The Verify3D was used to evaluate the 3D model’s quality. The 3D models scored 96.26% on this experiment’s evaluation scale, where the minimum passing score was 80% (Figure 2D).

### 2.3. Active Site Prediction

In the context of docking studies, it is imperative to pinpoint the putative binding pockets within the modeled protein structures. The 3D structure of RmAChE1 has not been documented in available databases, resulting in the absence of any reported catalytic sites. As a result, it is imperative to anticipate the potential binding regions of the receptor and determine the area with the largest cavity for docking compounds. For this purpose, we utilized the site finder tool integrated into MOE to determine the probable binding sites within RmAChE1. The analysis resulted in the identification of eight high-potential binding pockets. After evaluating the potential binding pockets, we selected the one that comprises the following amino acids: GLN122, VAL123, LEU124, ASP125, THR126, LEU127, SER134, TRP137, ASN138, ALA139, TYR173, GLY174, GLY175, GLY176, TYR178, Ser179, GLY180, THR181, LEU184, TYR187, GLU255, SER256, TRP289, THR335, ASN336, SER337, GLY338, GLY339, VAL340, VAL341, ASP342, PHE343, PRO344, TRP384, and PHE385. Additionally, utilizing the procedure reported by Empereur-Mot et al. (2015), a test of activity predictability was used to confirm the active site of the modeled protein. The aforementioned active sites were docked with active molecules [24]. Hence, as demonstrated in Figure 3, every active compound was precisely docked into the suggested active site, while inactive ones could not be docked at the intended active site (Figure 3). In the case of TIM, the catalytic site has been previously reported [25]. As such, we selected the corresponding residues at the catalytic site and performed docking simulations with the related compounds. Figure 3 demonstrates the active sites in the 3D structures of RmTIM and RmAChE1 (Figure 4).

### 2.4. Interaction Pattern of Virtual Hits

The interaction modes of the RmAChE1 and RmTIM proteins with flavonoid derivatives have been determined through PLIP https://plip-tool.biotec.tu-dresden.de (accessed on 27 Febuary 2023), an online tool for protein–ligand interactions. Table 1 shows the selected flavonoid molecules having maximal interaction scores with RmAChE1 and RmTIM.

Figure 5 illustrates the interaction mode of (A) methylenebisphloridzin, (B) thearubigin, and (C) fortunellin with RmAChE1. The methylenebisphloridzin molecule was found to create seven hydrogen bonds with specific amino acids (Gly 180, Trp 137, Tyr 388, Asp 125, Thr 126, Thr 135, and Gln 122) of the corresponding protein. The distances of the hydrogen bonds between amino acids and a different moiety of methylenebisphloridzin were measured to be 3.11, 1.75, 2.31, 2.99, 1.81, and 2.38Å. Methylenebisphloridzin also forms hydrophobic interactions between Tyr 178 and Asn 135. Thearubigin was found to interact with RmAChE1 through both hydrogen bonds and hydrophobic interactions. Thearubigin was observed to form hydrogen bonds with Asp 125, Val 123, Ala 208, and Asp 342 amino acids, with bond distances measuring 3.02, 2.66, 3.38, and 2.52 Å, respectively. Tyr 388 and Phe 389 show hydrophobic interaction with thearubigin. Fortunellin exhibited a similar interaction pattern as thearubigin, forming hydrogen bonds with Trp 384, Phe 389, Glu 255, and Gly 175 amino acids, with bond distances of 2.26, 2.57, 1.84, and 2.42 Å, respectively. Asp 125, Leu 127, and Tyr 388 show hydrophobic interaction with fortunellin.

Figure 6 indicates the interaction mode of (A) quercetagetin-7-*O*-(6-*O*-caffeoyl-β-d-glucopyranoside), (B) isorhamnetin, and (C) liquiritin towards the RmTIM protein. Quercetagetin interacts with Lysine 13 and Serine 96 amino acids by forming two hydrogen bonds at distances of 2.77 and 3.49 Å, respectively. In addition, hydrophobic interactions between quercetagetin and the amino acids Thr 172, Lys 174, Ser 211, Gly 233, His 15, and His 100 are also observed. Isorhamnetin forms three hydrogen bonds with Lys 13, Glu 96, and Leu 237 amino acids, with bond distances of 2.79, 2.41, and 2.85 Å, respectively. Additionally, isorhamnetin also shows hydrophobic interactions with Ile 170, Lys 274, Ser 211, and Val 231. Liquiritin is observed to interact with specific amino acid residues in its binding site through both hydrophobic contacts and hydrogen bonds. Specifically, it forms hydrophobic contacts with Lys 174, Ile 170, Glu 239, Ser 235, and Ala 234, as well as five hydrogen bonds with Lys 13, Glu 97, Ser 211, Asn 213, and Lys 237. These interactions occur at distances of approximately 2.81, 2.26, 2.47, 2.32, and 2.65 Å, respectively. The docking interactions of other selected flavonoids with RmAChE1 is shown in Appendix A.

### 2.5. Results of ADMET Calculation

Drug discovery and development is a complex process, and the most crucial yet challenging step is the conduct of DMPK (drug metabolism and pharmacokinetics) studies. This step accounts for the failure of approximately 60% of drugs during clinical phases. ADMET (absorption, distribution, metabolism, excretion, and toxicity) studies are a vital component of pharmacokinetics/pharmacology, describing the disposition of drug compounds in the body. The ADMET Predictor is a computer-designed program that utilizes molecular structures to estimate the pharmacokinetic properties/parameters of drug-like compounds, as outlined by [26].

The freely available Swiss ADME web tool is a software application that predicts the physicochemical properties, absorption, distribution, metabolism, elimination, and pharmacokinetic properties of molecules, all of which are critical determinants for clinical trials. This tool considers six physicochemical properties that are essential, including lipophilicity, flexibility, saturation, polarity, solubility, and size, as stated by [27].

The results of the ADMET evaluation provided insights into the physicochemical properties of the designed compounds, including compliance with the rules of five (MW, iLOGP, HBAs, and HBDs), as well as other parameters such as molecular polar surface area (TPSA), Blood-Brain Barrier (BBB) permeability, number of aromatic heavy atoms, and the presence of alerts for undesirable substructures (i.e., PAINS #alert), among others, and are presented in Table 2.

Table 2 presents several parameters of the designed compounds, including molecular weight (MW), number of rotatable bonds (RB), number of hydrogen donors (HBD), number of hydrogen acceptors (HBA), Topological Polar Surface Area (TPSA), octanol/water partition coefficient (iLOGP), number of aromatic heavy atoms (nAH), Molar refractivity (MR), and the number of alerts for undesirable substructures/substructures (PAINS #alert).

### 2.6. Density Functional Theory (DFT) Results

The frontier molecular orbitals HOMO and LUMO convey valuable data about any molecule’s reactivity. The difference in HOMO and LUMO energies (Eg) plays a significant role in comprehending the chemical reactivity and kinetic stability. It was a proportional function between the energy band gap and chemical reactivity and stability. For example, if the band gap in a molecule has a higher value, then the molecule is less polarizable and shows high kinetic stability and low chemical reactivity (hard molecule). The HOMO and LUMO of liquiritin, isorhamnetin, and complex were made. Energy details are given in Table 3, such as EHOMO, ELUMO, and Eg. Liquiritin and isorhamnetin possess band gaps of 0.16 eV and 0.005 eV, respectively, having the constituent energies of HOMO and LUMO, −0.33641 eV, −0.32355 eV and −0.20637 eV, −0.31813, respectively. The band gap shown by the complex was lower than the obtained band gaps of liquiritin, making it more reactive towards the affected area. While in the case of isorhamnetin, the band gap of complexes is more than isorhamnetin itself, making it stable reactive. This much smaller band gap of complex predicts the need for a small quantity of energy to get excited from ground level. Furthermore, this lesser energy difference shows the extent of conjugation within the LIQ and ISO (Table 4 and Table 5), whereas the Table 6 represents the HOMO-LUMO of selected flavonoids.

## 3. Discussion

Nowadays, synthetic insecticides are often employed to eliminate veterinary and pharmaceutical pests. Pest resistance, the product remains, active component withdrawal, environmental persistence, and nontarget dangers need new solutions. One approach uses plant-based chemicals’ toxic, irritant, or beneficial qualities. These compounds have limited human harm, short environmental persistence, and complicated chemistry, making them ideal pesticide candidates [28]. New bioactive compounds that are more potent and selective for tick targets have been explored as potential solutions to these problems [29]. In silico methods like molecular docking, homology modeling, and molecular dynamics make it easier to find novel compounds that bind to parasites’ molecular targets [30].

The neuronal enzyme AChE1, which catalyzes the breakdown of acetylcholine, is a recognized chemical target of ticks [31]. It causes neuromuscular paralysis and death by prolonging the tick’s neural excitation. Moreover, TIM is an enzyme involved in both glycolysis and gluconeogenesis since it catalyzes the conversion of 3-phosphoglycerate to dihydroxyacetone phosphate. To generate selective inhibitors, one must target the enzyme’s interface, which is poorly conserved between species, even if the structural similarity of the enzyme is well conserved across species [12]. However, whether a specific molecular inhibitor of RmTIM has a flavonoid structure or any other structure is still unknown.

Antiparasitic activity and enzyme inhibition are two important biological effects that piqued modern interest in the flavonoid family of natural secondary metabolites. Plant products’ acaricidal efficacy has been connected to the presence of flavonoids. Ghosh et al. discovered that adult *Ricinus microplus* might be destroyed in vitro by an ethanolic extract of Ricinus communis leaf containing flavonoids (quercetin, flavone, and kaempferol) [32]. In addition, the active fractions of Ocotea aciphylla leaf containing a flavonoid (vitexin-2″-*O*-rhamnoside 9) showed in vitro anti-tick efficacy against larvae of *R. microplus* [33]. Flavonoids’ capacity to bind to enzymes like AChE1 is another important biological characteristic. Researchers have shown that flavonoids (quercetin, rutin, narigin, and hesperidin) may inhibit AChE1 from *Electrophorus electricus* in vitro [34,35]. In silico and in vitro studies have shown scopoletin and diosmin to inhibit mouse AChE1 [36]. Despite this, there is a lack of research on plant flavonoids’ effect on the acaricidal effectiveness of RmAChE1 and RmTIM proteins.

Plant flavonoids’ docking with the RmAChE1 and RmTIM proteins was examined in this work. Docking analysis requires a protein’s 3D structure. Since the structure of the RmAChE1 protein has not been identified experimentally, we constructed a model of this protein using the homology modeling approach, widely accepted as the best technique for modeling proteins. The structure of RmTIM was obtained from the PDB. Multiple web-based tools assessed the accuracy and stereochemistry of the RmAChE1 model developed by the computer. Docking analysis requires a protein structure file in (.pdb) format. Docking analysis foretells a compound’s binding energy to a protein and the optimal molecular orientation for binding.

Our primary objective in this work was to determine whether natural flavonoids had any effect on the RmAChE1 and RmTIM proteins. This study retrieved 80 flavonoid structures from PubChem and PubMed. After that, these flavonoids were put through a virtual screening using MOE software, which tested them against both proteins. Ten flavonoids, including methylenebisphloridzin, thearubigin, fortunellin, quercetagetin-7-*O*-(6-*O*-caffeoyl-β-d-glucopyranoside) 1, quercetagetin-7-*O*-(6-*O*-*p*-coumaroyl-β-glucopyranoside), rutin, kaempferol 3-neohesperidoside, quercetagetin-7-*O*-(6-*O*-caffeoyl-β-d-glucopyranoside) 2, isorhamnetin, and liquiritin were obtained as potential inhibitors (the flavonoids 1–7 against RmAChE1; the flavonoids 8–10 against RmTIM) based on best docking scores against the active sites of both proteins of *R. microplus*. Many different pharmacological effects, such as antidiabetic, anti-inflammatory, anti-hyperglycemic, anti-cancer, antibacterial [37], AChE1 inhibitory [19], enzymatic [38], hepatoprotective [39], and antioxidant [40], may be applicable. Furthermore, we determined the drug-likeness properties using the available online tool SwissADME comprised of pharmacodynamics and pharmacokinetics and, ultimately, provided the prediction regarding absorption, distribution, metabolism, and excretion (ADME) in the human body like a drug. We incorporated the drug-like calculation based on the Lipinski rule of five to determine oral absorption or membrane permeability (Table 2). Following Lipinski’s rule of five and the concept of QED, as outlined in Table 2, all the designed compounds violated more than one rule, except Isorhamnetin and Liquirtin, which did not violate any rule. This implies that the MW, RB, HBD, HBA, TPSA, and iLOGP of some compounds fall within the acceptable range. Additionally, only one PAINS alert was identified for compounds 2, 4, 5, 6, and 8, indicating their specificity. Thus, it can be concluded that these active anti-tick compounds (1 to 10) possess an average pharmacokinetic profile.

The DFT results also revealed that both compounds (isorhamnetin and liquiritin) have better chemical reactivity and kinetic stability with considerable intramolecular charge transfer between electron-donor and electron-acceptor groups. The molecule with a higher band gap was less polarizable and showed high kinetic stability and low chemical reactivity (hard molecule). The band gap shown by the complex was lower than the obtained band gaps of liquiritin, which makes it more reactive towards the affected area. While in the case of isorhamnetin, the band gap of complexes was more than isorhamnetin itself, which makes it stable and reactive. This much smaller band gap of complex predicts the need for a small quantity of energy to get excited from ground level [41].

The stability and biological activity of flavonoids can be influenced by temperature, and their sensitivity to heat treatment can vary based on their chemical structure. Generally, glycosylated flavonoids exhibit more excellent heat treatment resistance than aglycone flavonoids [42]. Although the mechanism is unknown, incorporating acyl groups has been found to increase the thermostability of flavonoids [43]. For instance, Ishihara and Nakajima conducted a study where quercetin-3-glucoside was monoacylated in vitro using nine different aromatic carboxylic acids. The resulting flavonoids showed improved thermostability and light resistivity [44].

Based on our findings, flavonoids may function as inhibitors of the RmAChE1 and RmTIM proteins. Researchers studying plant flavonoids may gain new insight into their abilities to inhibit the RmAchE1 and RmTIM for treating *R. microplus* infection. Further research on flavonoid inhibition may lead to their use as acaricides in creating novel antiparasitic medications.

## 4. Method and Materials

### 4.1. Preparation of TIM and the Homology Model of RmAChE1

TIM’s 3D structure [PDB: 3TH6] was retrieved from the PDB archive of the Research Collaboratory for Structural Bioinformatics (RCSB) (https://www.rcsb.org (accessed on 10 June 2022)). To prepare the TIM protein structure for molecular docking, crystal water molecules were eliminated. Some residues were missing in the structure of TIMs, so we constructed the missing residues using a loop modeler implemented in MOE software.

Since the 3D structure of the RmAChE1 protein was not reported, we needed to model the structure of RmAChE1. The amino acid sequence of RmAChE1 was retrieved from uniport https://www.uniprot.org/uniprotkb/ (accessed on 11 June 2022) under accession No. A0A0F6P2D6 RHIMP and then modeled using SWISS-MODEL https://swissmodel.expasy.org/interactive/ (accessed on 12 June 2022). To identify appropriate templates for constructing the 3D structure of RmAChE1, a sequence similarity search was carried out using the NCBI BLAST against the PDB database. Sequence alignment analysis was conducted through the Clustal Omega server [45]. In addition, the stereochemical quality of the modeled structure of the test protein was validated using the PROCHECK, Verify3D, and ERRAT services.

### 4.2. Prediction of the Active Site

RmAChE1 lacks a determined 3D structure, and thus its active site has not been experimentally identified. Therefore, we utilize site finder, a computational tool implemented in the molecular operating environment, to evaluate the potential active site. These site prediction tools provide a theoretical assessment of the active site in RmAChE1. In addition, TIM is a dimeric protein comprising two individual monomers, each containing a TIM barrel structure. The active site responsible for the catalytic activity is situated within this TIM barrel. Given the high similarity of the catalytic sites among all TIM structures, a set of catalytic residues was selected for further investigation through molecular docking studies.

### 4.3. Ligands Searching and Database Preparation

An extensive literature survey of flavonoids revealing the AChE inhibiting potential [46], easy availability, and most structures available in the PubChem database served as the rationale for the flavonoid selection for this study. Flavonoids’ chemical structures were obtained from PubMed (https://www.ncbi.nlm.nih.gov/pubmed (accessed on 15 June 2022)), PubMed Central (https://www.ncbi.nlm.nih.gov/pmc (accessed on 15 June 2022)), and PubChem (https://www.ncbi.nlm.nih.gov/pccompound (accessed on 15 June 2022)) as shown in Appendix A. The structures were saved as (.mol) files. Atoms of hydrogen were incorporated into the structural optimization conducted with MOE software. We could use these characteristics to reduce the energy of specific molecules (gradient: 0.05, force field: the Merck molecular force field 94× (MMFF94×), chiral constraint, and current geometry). The revised flavonoid structures were then incorporated into the molecular docking database (.mdb) for docking analysis.

### 4.4. Docking Analysis

Docking analysis was performed using the MOE software. A docking study was conducted using the best available AChE1 and TIM structures via the SWISS-MODEL tool from the PDB. Energy minimization and 3D protonation were performed on the protein structure using the following settings (force field: MMFF94× solvation, chiral constraint, current geometry, gradient: 0.05). These simplified structures were then used as receptors in docking protocols. MOE’s site discovery module was crucial in locating a protein’s active site. The docking process was carried out using MOE’s default configurations. The docking score of the MOE software was used to figure out the binding free energies of each ligand structure in a given orientation.

### 4.5. Ligand Interaction Calculations

The ligand interactions were calculated using the MOE Program’s Ligand Interaction module (https://www.chemcomp.com/Products.htm (accessed on 25 June 2022)). It combines predicted distances between interacting atoms from 2D and 3D ligand–receptor protein models.

### 4.6. ADMET and Drug-Likeness Evaluation

To ensure the safety and efficacy of drug candidates, it is crucial that they exhibit desirable ADME properties and are non-toxic. Consequently, an assessment was conducted on the ADME profile of the synthesized compounds, which involved the evaluation of drug-likeness, partition coefficient, solubility, and various other parameters using the SwissADME module [26]. This module is available on the SIB (Swiss Institute of Bioinformatics) webserver at https://www.sib.swiss (accessed on 2 July 2022).

### 4.7. DFT Analysis

DFT analysis of compounds was performed using Gaussian 09 software and visualized through Gauss view 5.0. The structural coordinates of the lead compounds were optimized using B3LYP/6-31 G (d,p) level basis sets without any symmetrical constraints. Frontier molecular orbitals analysis was applied for selected flavonoid molecules.

## 5. Conclusions

Our research elucidated the structure and interactions of cattle tick proteins with plant flavonoids. In the last decade, flavonoids have been the subject of much research. Their potential health benefits have also been explained beyond their nutritional value. Newly discovered 3D structures of RmTIM and RmAChE1 have acceptable stereochemical and energetic qualities, allowing for the prediction of novel inhibitors. This research aims to evaluate the efficacy of flavonoids from medicinal plants as acaricides against proteins produced by the parasite *R. microplus*. The seven flavonoids (methylenebisphloridzin, thearubigin, fortunellin, quercetagetin-7-*O*-(6-*O*-caffeoyl-β-d-glucopyranoside) 1, quercetagetin-7-*O*-(6-*O*-*p*-coumaroyl-β-glucopyranoside), rutin, and kaempferol 3-neohesperidoside) were docked against RmAChE1, proving their strong interaction. This work will help make and evaluate flavonoids as tick acaricides to prevent tick-borne illnesses.

## Figures and Tables

**Figure 1 molecules-28-03606-f001:**
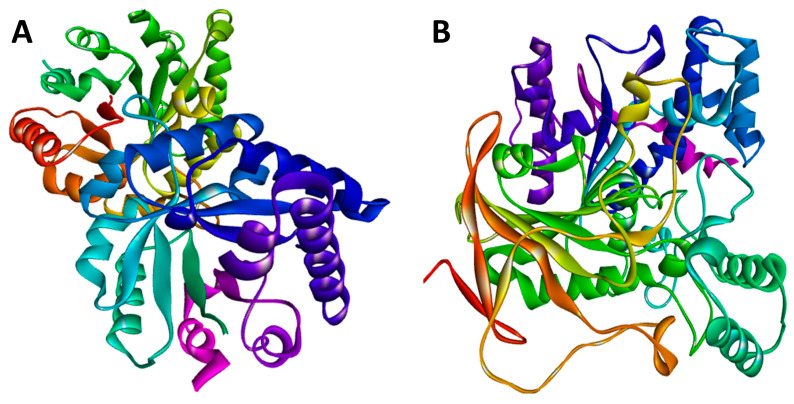
The structural models were predicted using the online SWISS-MODEL tool. (**A**) The three-dimensional (3D) structure for the triose-phosphate isomerase (TIM) of *R. microplus* (RmTIM); (**B**) The 3D structure for the acetylcholinesterase (AChE1) of *R. microplus* (RmAChE1).

**Figure 2 molecules-28-03606-f002:**
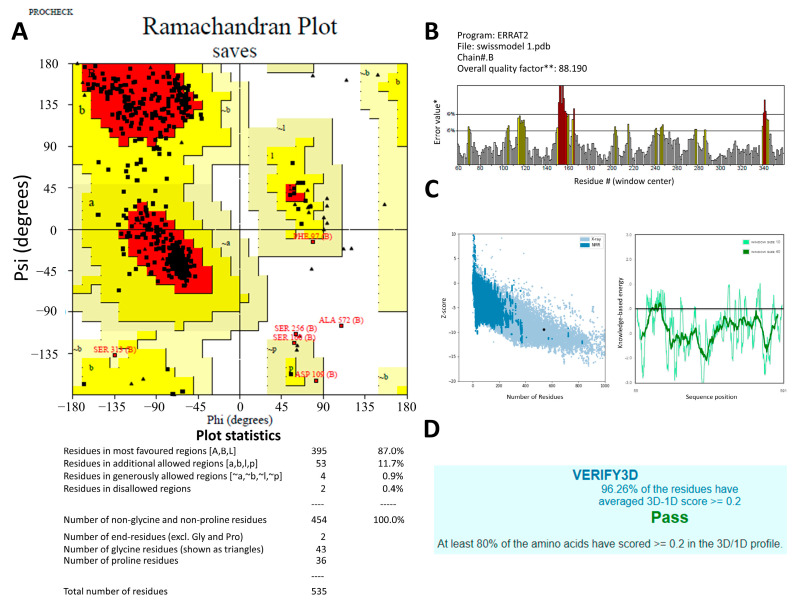
The predicted 3D structures of the Rhipicephalus microplus acetylcholinesterase 1 (RmAChE1) protein were verified using (**A**) the Ramachandran plot produced by PROCHECK, where the red, yellow and black colors represent most favorable, favorable, and disallowed region respectively, Phi and Psi bond represent torsion angle which predict the possible conformation of the peptides. (**B**) the ERRAT server quality factor, * On the error axis, two lines are drawn to indicate the confidence with which it is possible to reject regions that exceed that error value. ** Expressed as the percentage of the protein for which the calculated error value falls below the 95% rejection limit. Good high-resolution structures generally produce values around 95% or higher. For lower resolutions (2.5 to 3 Å) the average overall quality factor is around 91%. The red and yellow bars represent the 99% and 95% confidence respectively. (**C**) the ProSA-web (Z-score), and (**D**) the Verify3D.

**Figure 3 molecules-28-03606-f003:**
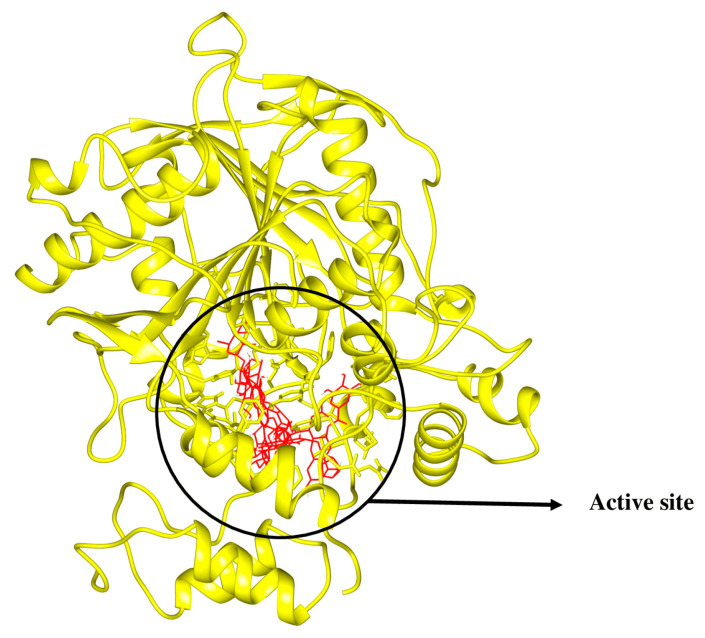
Active compounds were docking in the mentioned active sites colored in red.

**Figure 4 molecules-28-03606-f004:**
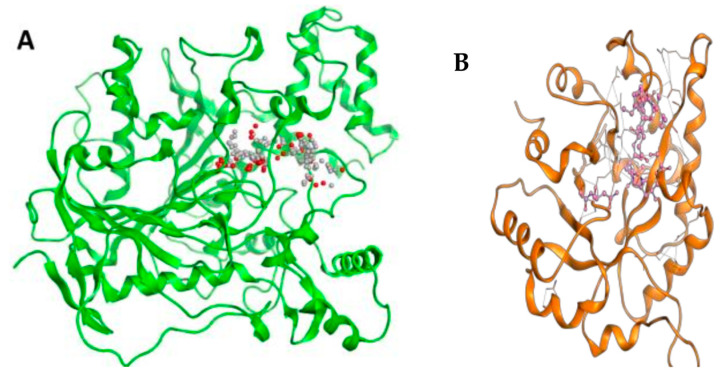
(**A**) RmAChE1 3D structure where active sites are represented by small red and grey dots and (**B**) RmTIM’s 3D structure where the colored portion in center shows active sites.

**Figure 5 molecules-28-03606-f005:**
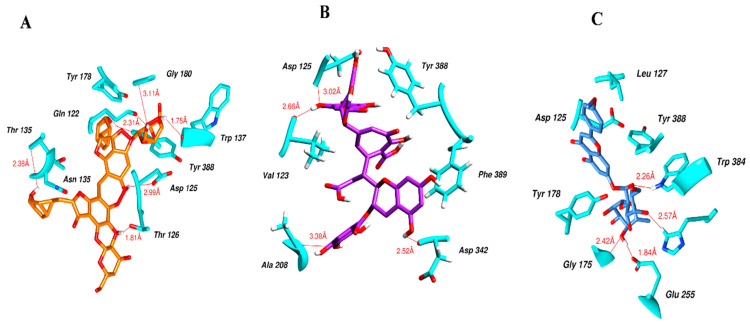
Docking interactions between (**A**) methylenebisphloridzin, (**B**) thearubigin, and (**C**) fortunellin and the RmAChE1 protein. The center part of these subfigures represent the ligand surrounded by the interacting amino acids.

**Figure 6 molecules-28-03606-f006:**
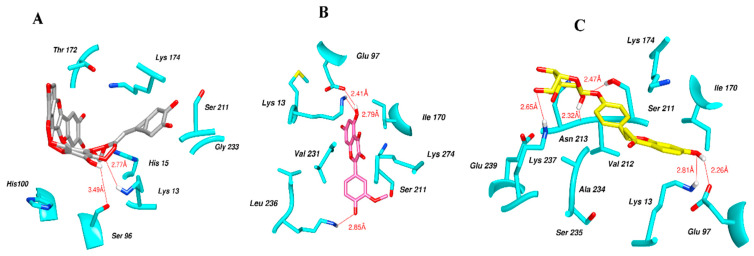
Docking interactions between (**A**) quercetagetin-7-*O*-(6-*O*-caffeoyl-β-d-glucopyranoside) 2, (**B**) isorhamnetin, and (**C**) liquiritin and the RmTIM protein. The center part of these subfigures represent the ligand surrounded by the interacting amino acids.

**Table 1 molecules-28-03606-t001:** Selected flavonoid molecules downloaded from PubChem and PubMed with maximal docking scores.

No.	Name	PubChem CID	IUPAC Names	Compound Structures	Plant Source	Docking Scores (Kcal/mol)	Targeted Protein
1.	Methylenebisphloridzin	-	-	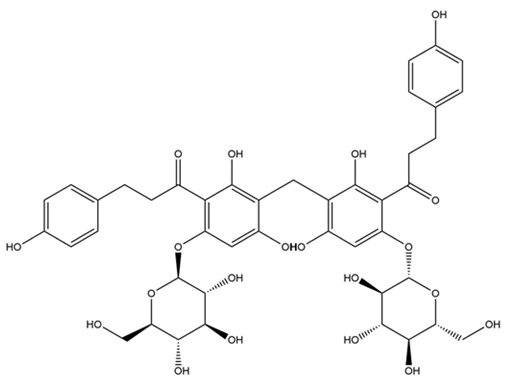	Eupatorium adenophorum	−11.0455	Acetylcholinesterase of *Rhipicephalus microplus* (RmAChE1)
2.	Thearubigin	100945367	7-[2-carboxy-1-[(2*R*,3*R*)-5,7-dihydroxy-3-(3,4,5-trihydroxybenzoyl)oxy-3,4-dihydro-2H-chromen-2-yl]ethyl]-5-[(2*R*,3*R*)-5,7-dihydroxy-3-(3,4,5-trihydroxybenzoyl)oxy-3,4-dihydro-2H-chromen-2-yl]-2-hydroxy-3-oxocyclohepta-1,4,6-triene-1-carboxylic acid	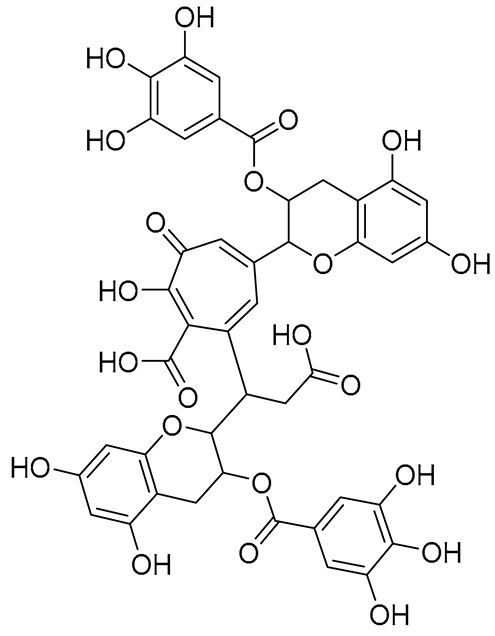	Camellia sinensis, Camellia assamica	−9.0071	RmAChE1
3.	Fortunellin	5317385	7-[(2*S*,3*R*,4*S*,5*S*,6*R*)-4,5-dihydroxy-6-(hydroxymethyl)-3-[(2*S*,3*R*,4*R*,5*R*,6*S*)-3,4,5-trihydroxy-6-methyloxan-2-yl]oxyoxan-2-yl]oxy-5-hydroxy-2-(4-methoxyphenyl)chromen-4-one	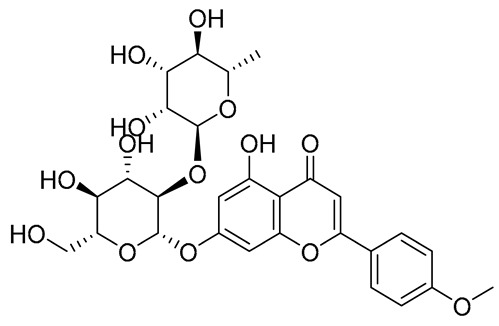	Fortunella japonica, Fortunella margarita, Fortunella crassifolia, and Fortunella hindsii	−8.9384	RmAChE1
4.	Quercetagetin-7-*O*-(6-*O*-caffeoyl-β-d-glucopyranoside) 1	-	-	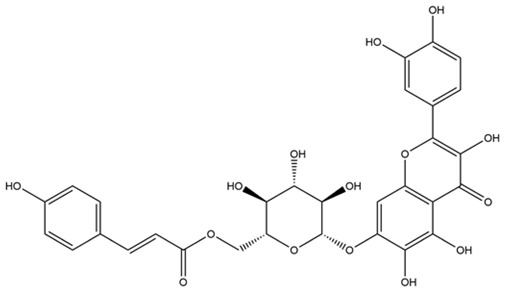	Eupatorium adenophorum	−8.6203	RmAChE1
5.	Quercetagetin-7-*O*-(6-*O*-*p*-coumaroyl-β-glucopyranoside)	-	-	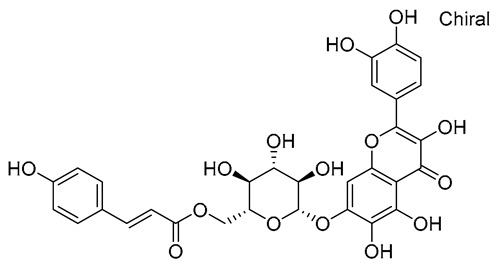	Eupatorium adenophorum	−8.5624	RmAChE1
6.	Rutin	5280805	2-(3,4-dihydroxyphenyl)-5,7-dihydroxy-3-[(2*S*,3*R*,4*S*,5*S*,6*R*)-3,4,5-trihydroxy-6-[[(2*R*,3*R*,4*R*,5*R*,6*S*)-3,4,5-trihydroxy-6-methyloxan-2-yl]oxymethyl]oxan-2-yl]oxychromen-4-one	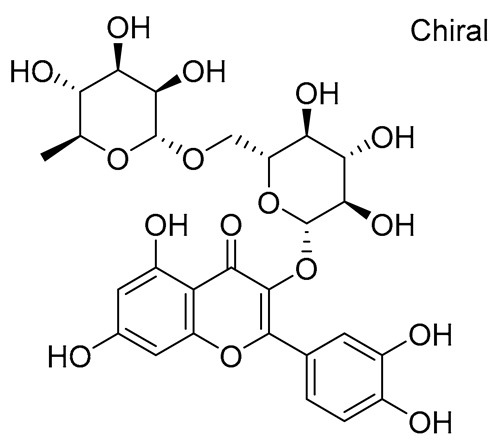	Ruta graveolens and Morus alba	−8.5489	RmAChE1
7.	Kaempferol 3-neohesperidoside	531876	3-[(2*S*,3*R*,4*S*,5*S*,6*R*)-4,5-dihydroxy-6-(hydroxymethyl)-3-[(2*S*,3*R*,4*R*,5*R*,6*S*)-3,4,5-trihydroxy-6-methyloxan-2-yl]oxyoxan-2-yl]oxy-5,7-dihydroxy-2-(4-hydroxyphenyl)chromen-4-one	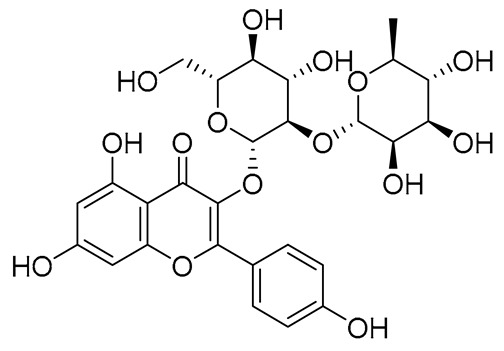	Daphniphyllum calycinum, Amygdalus persica	−8.2550	RmAChE1
8.	Quercetagetin-7-*O*-(6-*O*-caffeoyl-β-d-glucopyranoside) 2	-	quercetagetin-7-*O*-(6-*O*-caffeoyl-β-d-glucopyranoside	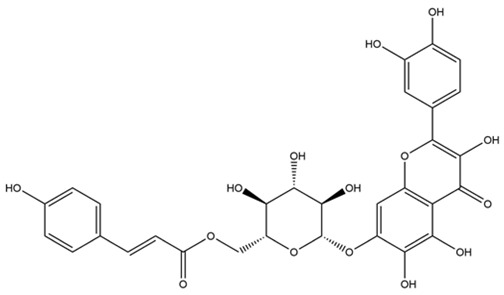	Eupatorium adenophorum	−9.8855	Triose-phosphate isomerase of *R. microplus* (RmTIM)
9.	Isorhamnetin	5281654	3,5,7-trihydroxy-2-(4-hydroxy-3-methoxyphenyl)chromen-4-one	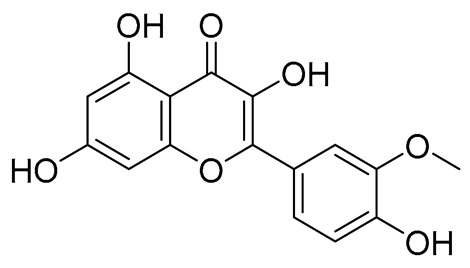	Calendula officinalis	−7.6324	RmTIM
10.	Liquiritin	503737	(2*S*)-7-hydroxy-2-[4-[(2*S*,3*R*,4*S*,5*S*,6*R*)-3,4,5-trihydroxy-6-(hydroxymethyl)oxan-2-yl]oxyphenyl]-2,3-dihydrochromen-4-one	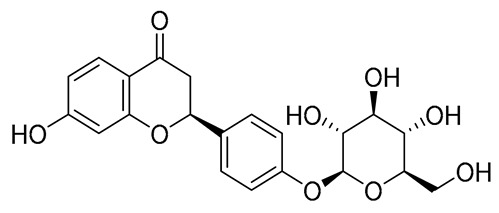	Glycyrrhiza glabra	−7.5141	RmTIM

**Table 2 molecules-28-03606-t002:** Physicochemical properties of flavonoid molecules calculated by SwissADME web tool.

No.	Compounds Name	MW (g/mol)	Log P	HBD	HBA	TPSA	No. of Violations	BBB	PAINS Alerts
1.	Methylenebisphloridzn	884.83	1.18	14	20	354.28	3	No	0
2.	Thearubigin	902.72	0.56	13	22	385.26	3	No	1
3.	Fortunellin	592.55	2.13	7	14	217.97	3	No	0
4.	Quercetagetin-7-*O*-(6-*O*-caffeoyl-β-d-glucopyranoside) 1	788.66	1.42	12	20	336.19	3	No	1
5.	Quercetagetin-7-*O*-(6-*O*-*p*-coumaroyl-β-glucopyranoside)	626.52	2.76	9	15	257.04	3	No	1
6.	Rutin	610.52	0.46	16	16	269.43	3	No	1
7.	Kaempferol 3-neohesperidoside	594.52	0.96	9	15	249.2	3	No	0
8.	Quercetagetin-7-*O*-(6-*O*-caffeoyl-β-d-glucopyranoside) 2	788.66	1.42	12	20	336.19	3	No	1
9.	Isorhamnetin	316.26	2.35	4	7	120.36	0	No	0
10.	Liquiritin	418.39	1.46	5	9	145.91	0	No	0

MW: Molecular Weight, LogP: Logarithm of the Partition Coefficient (P), HD: Hydrogen Bond Donor, HA: Hydrogen Bond Acceptor, TPSA: Topological Polar Surface Area, RBs: Number of Rotatable Bonds, BBB: Blood Brain Barrier.

**Table 3 molecules-28-03606-t003:** Frontier molecular orbitals such as HOMO and LUMO selected flavonoid molecules.

Description	eV HOMO	eV LUMO	Energy Gap
* Liquiritin	−0.33641	−0.20637	0.15773
** Isorhamnetin	−0.32355	−0.31813	0.00542
*** ISOalanine	−0.27184	−0.20810	0.06374
*** ISOproline	−0.27471	−0.21139	0.06332
*** LIQphenylalanine	−0.28800	−0.20662	0.08138
*** LIQleucine	−0.29611	−0.20819	0.08792
*** LIQlysine	−0.29590	−0.20821	0.08769

* Most Stable; ** Most Reactive; *** Stable Reactive/Effective.

**Table 4 molecules-28-03606-t004:** (**A**) Isorhamnetin 3D structure, HOMO-LUMO, and band gap; (**B**) 3D structure of the isorhamnetin complex with proline amino acid, HOMO-LUMO, and band gap; (**C**) 3D structure of the isorhamnetin complex with alanine amino acid, HOMO-LUMO, and band gap.

S. No.	3D-Structures	HOMO-LUMO	Band Gap
(**A**)	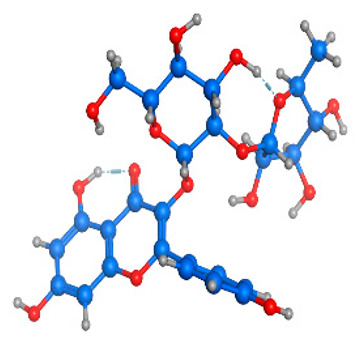	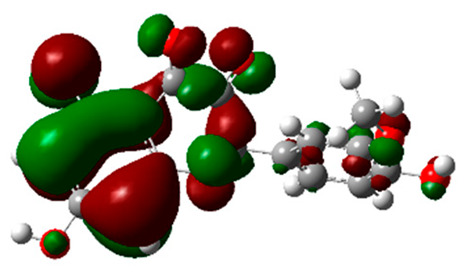	0.00542
(**B**)	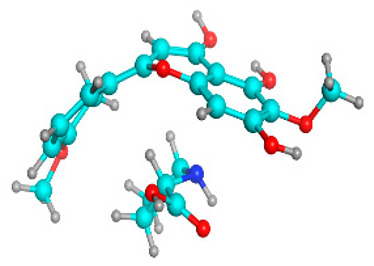	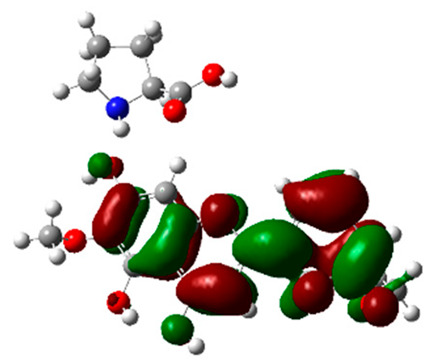	0.06332
(**C**)	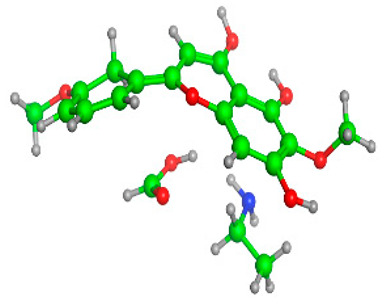	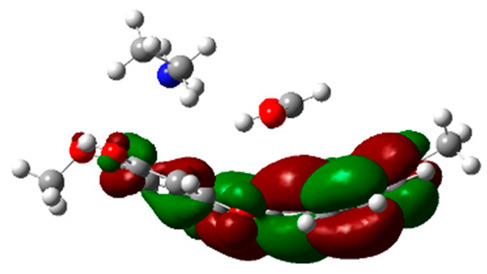	0.06374

**Table 5 molecules-28-03606-t005:** (**A**) Liquiritin 3D structure, HOMO-LUMO, and band gap; (**B**) 3D structure of liquiritin complex with leucine amino acid, HOMO-LUMO, and band gap; (**C**) 3D structure of liquiritin complex with lysine amino acid, HOMO-LUMO, and band gap; (**D**) 3D structure of liquiritin complex with phenylalanine amino acid, HOMO-LUMO, and band gap.

S. No.	3D-Structures	HOMO-LUMO	Energies
(**A**)	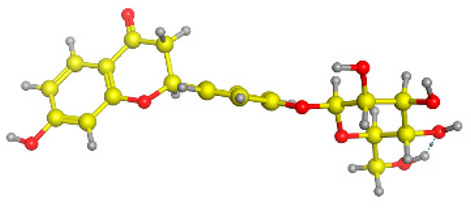	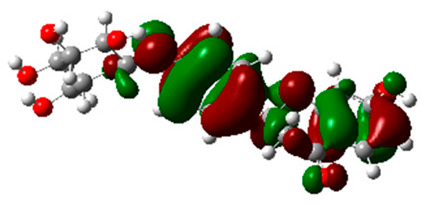	0.15773
(**B**)	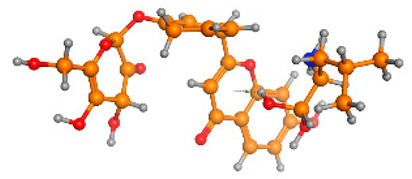	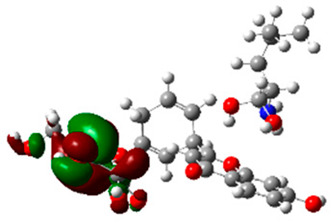	0.08792
(**C**)	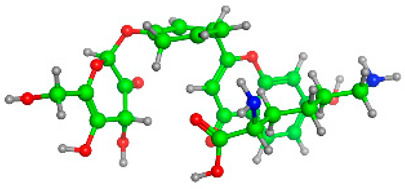	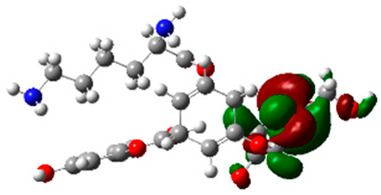	0.08769
(**D**)	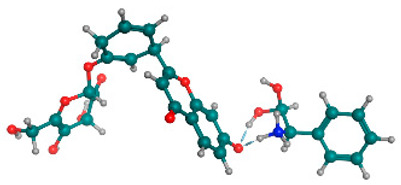	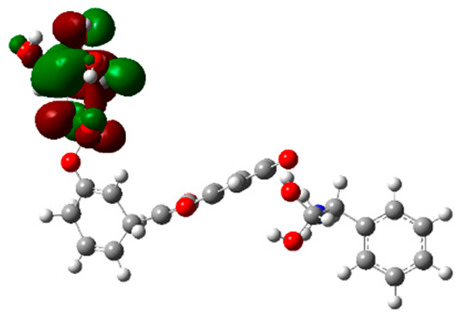	0.08138

**Table 6 molecules-28-03606-t006:** Selected flavonoid molecules with 3D Structure and HOMO-LOMO.

No.	Compound Name	3D Structure	HOMO-LOMO
1.	Methylenebisphloridzin	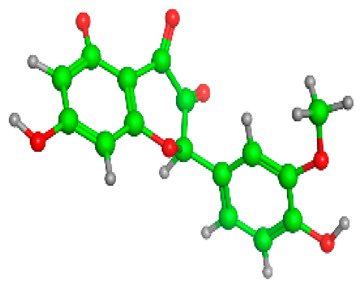	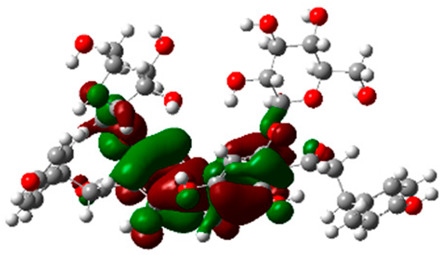
2.	Thearubigin	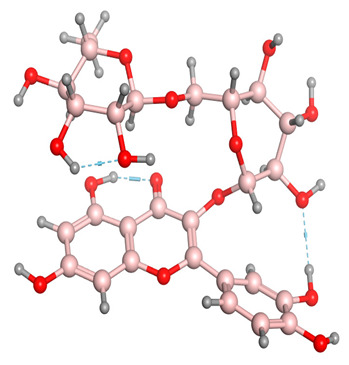	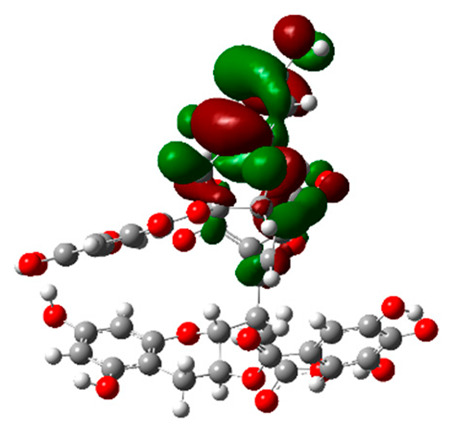
3.	Fortunellin	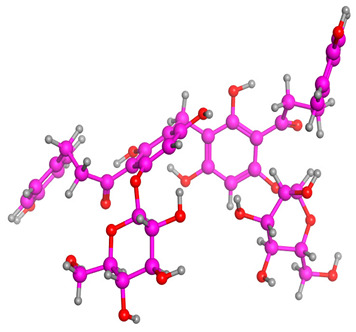	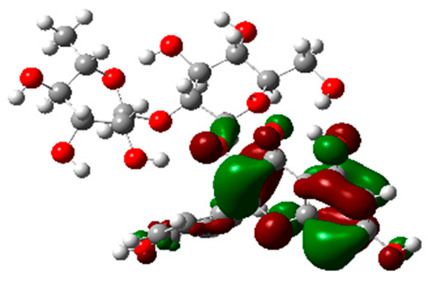
4.	Quercetagetin-7-*O*-(6-*O*-caffeoyl-β-d-glucopyranoside) 1	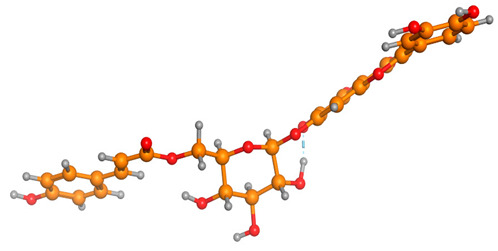	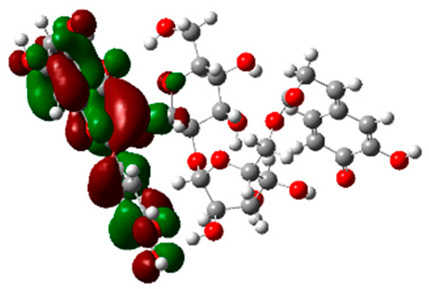
5.	Quercetagetin-7-*O*-(6-*O*-*p*-coumaroyl-β-glucopyranoside)	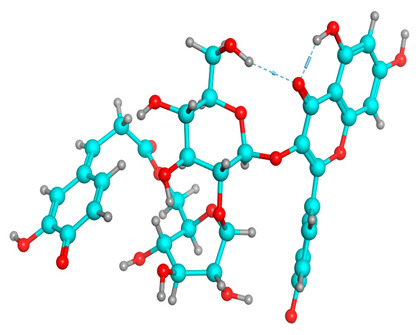	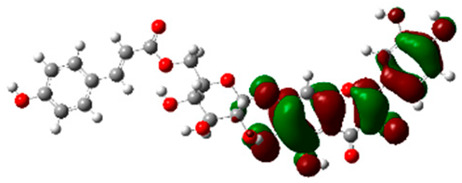
6.	Rutin	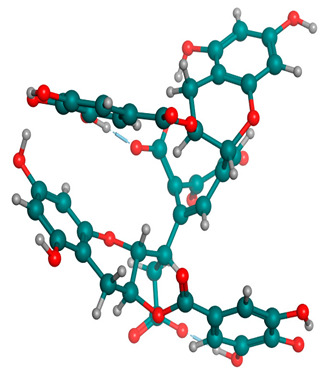	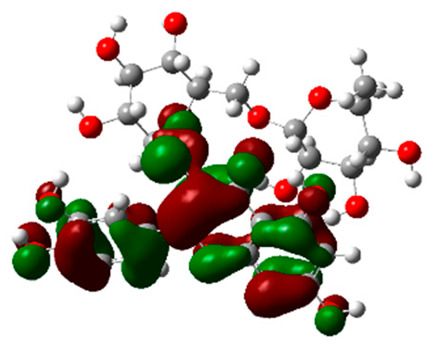
7.	Kaempferol 3-neohesperidoside	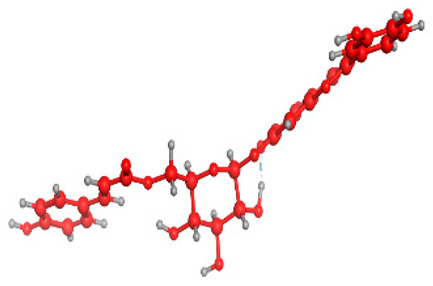	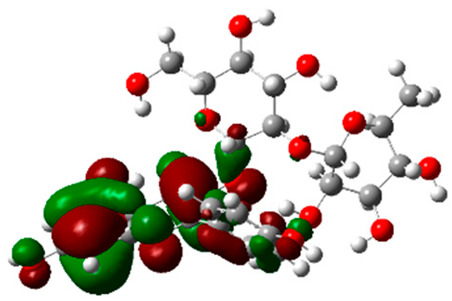
8.	Quercetagetin-7-*O*-(6-*O*-caffeoyl-β-d-glucopyranoside) 2	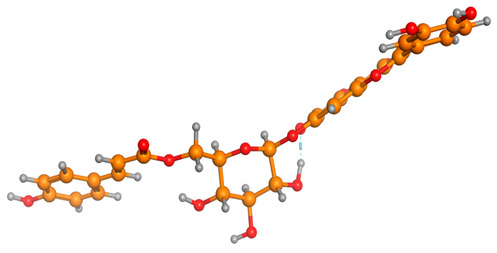	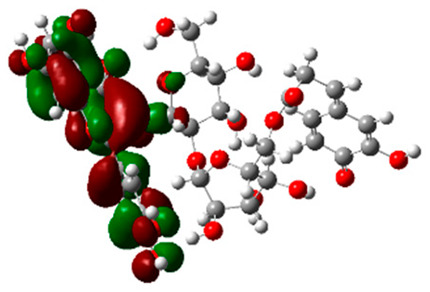
9.	Isorhamnetin	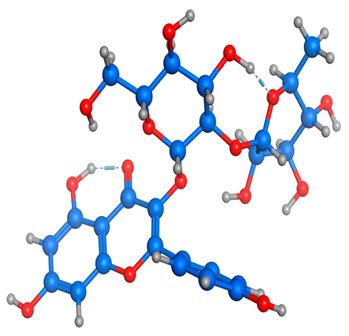	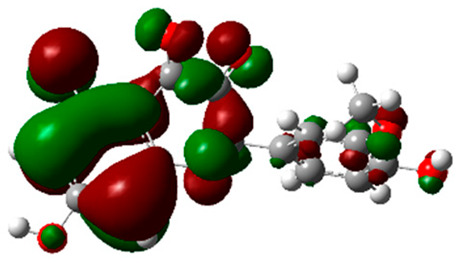
10.	Liquiritin	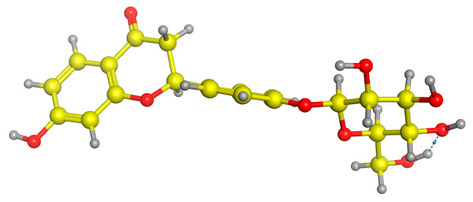	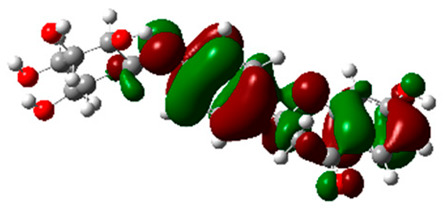

## Data Availability

The data supporting this study’s findings are available from the corresponding authors upon reasonable request.

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
