# Peer review of "Density Functional Theory Calculations and Molecular Docking Analyses of Flavonoids for Their Possible Application against the Acetylcholinesterase and Triose-Phosphate Isomerase Proteins of Rhipicephalus microplus"

_molecules, 2023, doi:10.3390/molecules28083606_

Round 1

Reviewer 1 Report

      This manuscript described flavonoids’ inhibitory effects on the acetylcholinesterase and triose-phosphate Isomerase proteins of Rhipicephalus microplus by molecular docking analyses. The manuscript is overall sound, with minor caveats as noted below:

1.      3.5. Molecular docking analysis

If possible, ten compounds with higher docking scores should be properly discussed with the other compounds. How do these ten compounds differ from other compounds in terms of chemical structure?

2.      3.6. Calculation of ligand interaction

Please discuss the differences among the RmAChE1 flavonoid-derived potent inhibitors as well as RmTIM inhibitors in terms of binding sites as appropriate in this section.

3.      The article points out that the"These computationally-driven discoveries are beneficial and can be utilized in assessing drug bioavailability in both in vitro and in vivo settings. " It is suggested to include in vitro toxicological studies of flavonoids 1-7 against RmAChE1 / flavonoids 8-10 against RmTIM to better verify the accuracy of these predicted results.

4.      The chemical structures of compounds listed in the table are not in a unified format. Please redraw them with the structure drawing tool. 

Author Response

Reviewer 1:

This manuscript described flavonoids' inhibitory effects on the acetylcholinesterase and triose-phosphate Isomerase proteins of Rhipicephalus microplus by molecular docking analyses. The manuscript is overall sound, with minor caveats as noted below:

->Thank you for your kind words and professional comments. We did our best to revise our draft point to point according to your comments.

1. 3.5. Molecular docking analysis. If possible, ten compounds with higher docking scores should be properly discussed with the other compounds. How do these ten compounds differ from other compounds in terms of chemical structure?

->Our docking algorithms compute a docking score related to the free energy of binding a ligand to a receptor. Based on scoring functions, we predicted the binding affinity between molecules after they had been docked. For this type of docking score, the more negative the score, the better the compounds are. The criteria of chemical structures were not considered while selecting these compounds for docking analysis. Instead, these compounds were selected based on their biological activities against parasites and insects from the literature.

2. 3.6. Calculation of ligand interaction. Please discuss the differences among the RmAChE1 flavonoid-derived potent inhibitors as well as RmTIM inhibitors in terms of binding sites as appropriate in this section.

->The interaction modes of the RmAChE1 and TIM protein with flavonoid derivatives have been determined through PLIP https://plip-tool.biotec.tu-dresden.de an online tool for protein-ligand interactions. Figure 1 illustrates the interaction mode of  (A) methylenebisphloridzin, (B) thearubigin, and (C) fortunellin with RmAChE1. The methylenebisphloridzin molecule was found to create seven hydrogen bonds with specific amino acids (Gly 180, Trp 137, Tyr 388, Asp 125, Thr 126, Thr 135, and Gln 122) of the corresponding protein. The distances of the hydrogen bonds between amino acids and dimoietiet moiety of methylenebisphloridzin were measured to be 3.11, 1.75, 2.31, 2.99, 1.81, and 2.38Å. Methylenebisphloridzin also forms hydrophobic interactions Tyr 178 and Asn 135. Thearubigin was found to interact with RmAChE1 through both hydrogen bonds and hydrophobic interactions. Thearubigin was observed to form hydrogen bonds with Asp 125, Val 123, Ala 208, and Asp 342 amino acids, with bond distances measuring 3.02, 2.66, 3.38, and 2.52 Å. Tyr 388 and Phe 389 show hydrophobic interaction with thearubigin. Fortunellin exhibited a similar interaction pattern as thearubigin, as it formed hydrogen bonds with Trp 384, Phe 389, Glu 255, and Gly 175 amino acids, with bond distances of 2.26, 2.57, 1.84, and 2.42 Å. Asp 125, Leu 127, and Tyr 388 show hydrophobic interaction with fortunellin.

The results depicted in Figure 2 indicate the interaction mode of A) quercetagetin-7-O-(6-O-caffeoyl-β-D-glucopyranoside), B) isorhamnetin, and C) liquiritin towards the RmTIM protein. Quercetagetin interacts with Lysine 13 and Serine 96 amino acids by forming two hydrogen bonds at distances of 2.77 and 3.49 Å, respectively. In addition, hydrophobic interactions between quercetagetin and the amino acids Thr 172, Lys 174, Ser 211, Gly 233, His 15, and His 100 are also observed. Isorhamnetin forms three hydrogen bonds with Lys 13, Glu 96, and Leu 237 amino acids, with bond distances of 2.79, 2.41, and 2.85 Å, respectively. Additionally, isorhamnetin also shows hydrophobic interactions with Ile 170, Lys 274, Ser 211, and Val 231. Liquiritin is observed to interact with specific amino acid residues in its binding site through both hydrophobic contacts and hydrogen bonds. Specifically, it forms hydrophobic contacts with Lys 174, Ile 170, Glu 239, Ser 235, and Ala 234, as well as five hydrogen bonds with Lys 13, Glu 97, Ser 211, Asn 213, and Lys 237. These interactions occur at distances of approximately 2.81, 2.26, 2.47, 2.32, and 2.65 Å.

3. The article points out that the"These computationally-driven discoveries are beneficial and can be utilized in assessing drug bioavailability in both in vitro and in vivo settings. " It is suggested to include in vitro toxicological studies of flavonoids 1-7 against RmAChE1 / flavonoids 8-10 against RmTIM to better verify the accuracy of these predicted results.

->Yes, I agree with your addressing points based on computationally-driven discoveries for drug bioavailability in both in vitro and in vivo settings. In vitro studies is another ongoing project to assess the toxicological studies of flavonoids 1-7 against RmAChE1 / flavonoids 8-10 against RmTIM to verify the accuracy of these predicted results better.

4. The chemical structures of compounds listed in the table are not in a unified format. Please redraw them with the structure drawing tool.

->Thanks for your comments. All chemical structures of compounds listed in the table are now provided in a unified format, as mentioned below.

S.No.

Compounds Name

Structures

1

Meth-ylenebisphloridzin

2

Thearubigin

3

Fortunellin

4

Quercetagetin-7-O-(6-O-caffeoyl-β-D-glucopyranoside) 1

5

Quercetagetin-7-O-(6-O-p-coumaroyl-β-glucopyranoside)

6

Rutin

7

Kaempferol 3-neohesperidoside

8

Quercetagetin-7-O-(6-O-caffeoyl-β-D-glucopyranoside) 2

9

Isorhamnetin

10

Liquiritin

Reviewer 2 Report

The authors reported seven flavonoids as potential inhibitors of rmAChE1 and three flavonoids as potential inhibitors of rmTIM via molecular docking and DFT calculations. These flavonoids might be served as eco-friendly strategies for managing ticks and tick-borne diseases.

Here, I have several questions and suggestions for the authors:

1, In the introduction part, could the authors provide us with more information about the experimental activities of the flavonoids on the AChE1s and TIMs of other species?

2, In the method part, could the authors offer more details about the binding-site residues minimization, which is crucial to molecular docking?

3, Is there any other reasonable docking poses for these flavonoids in these two targets? The docking scores might not be totally reliable for picking up the best poses.

4, The docking poses in Figures 4 and 5 need to be clarified. Could the authors highlight the interactions in the figures?

5, Could the authors discuss the stability of these flavonoids in the application?

Author Response

Reviewer 2:

The authors reported seven flavonoids as potential inhibitors of rmAChE1 and three flavonoids as potential inhibitors of rmTIM via molecular docking and DFT calculations. These flavonoids might be served as eco-friendly strategies for managing ticks and tick-borne diseases. Here, I have several questions and suggestions for the authors:

->Thank you for your professional summary and comments. We will do our bests to revise our draft point to point according to your comments.

1. In the introduction part, could the authors provide us with more information about the experimental activities of the flavonoids on the AChE1s and TIMs of other species?

->Literature has briefly mentioned the activities of flavonoids on AChE and BChE enzymes from various species, including humans, and their potential as anti-Alzheimer agents. However, there is no mention of flavonoid activity on TIMs of other species. Therefore, the comments have been incorporated in the introduction of the revised manuscript.

2. In the method part, could the authors offer more details about the binding-site residues minimization, which is crucial to molecular docking?

->In the context of docking studies, it is imperative to pinpoint the putative binding pockets within the modeled protein structures. The three-dimensional structure of RmAChE1 has not been documented in available databases, resulting in the absence of any reported catalytic sites. As a result, it is imperative to anticipate the potential binding regions of the receptor and determine the area with the largest cavity for the purpose of docking compounds. For this purpose, we utilized the site finder tool integrated into MOE to determine the probable binding sites within RmAChE1. The analysis resulted in the identification of eight high-potential binding pockets. After evaluating the potential binding pockets, we selected the one that comprises the following amino acids: GLN122, VAL123, LEU124, ASP125, THR126, LEU127, SER134, TRP137, ASN138, ALA139, TYR173, GLY174, GLY175, GLY176, TYR178, Ser179, GLY180, THR181, LEU184, TYR187, GLU255, SER256, TRP289, THR335, ASN336, SER337, GLY338, GLY339, VAL340, VAL341, ASP342, PHE343, PRO344, TRP384, and PHE385. In the case of TIM, the catalytic site has been previously reported (Marsh and Shah 2014). As such, we selected the corresponding residues at the catalytic site and performed docking simulations with the related compounds.

3. Is there any other reasonable docking poses for these flavonoids in these two targets? The docking scores might not be totally reliable for picking up the best poses.

->It is possible that there could be other reasonable docking poses for these flavonoids in the RmAChE1 and RmTIM targets. While the docking scores are often used to rank the binding affinities of different ligands, they may not always be entirely reliable for identifying the best binding pose. To increase confidence in the docking results, it is common practice to analyze multiple docking poses and select the most plausible ones based on various criteria, such as predicted binding energies, interactions with key residues, and consistency with experimental data.

4. Figures 4 and 5 need to be clarified. Could the authors highlight the interactions in the figures?

-> Our revisions are as follows:

3.4 Interaction Pattern of Virtual Hits

The interaction modes of the RmAChE1 and TIM protein with flavonoid derivatives have been determined through PLIP https://plip-tool.biotec.tu-dresden.de an online tool for protein-ligand interactions. Figure 4 illustrates the interaction mode of  (A) methylenebisphloridzin, (B) thearubigin, and (C) fortunellin with RmAChE1. The methylenebisphloridzin molecule was found to create seven hydrogen bonds with specific amino acids (Gly 180, Trp 137, Tyr 388, Asp 125, Thr 126, Thr 135, and Gln 122) of the corresponding protein. The distances of the hydrogen bonds between amino acids and a different moiety of methylenebisphloridzin were measured to be 3.11, 1.75, 2.31, 2.99, 1.81, and 2.38Å. Methylenebisphloridzin also forms hydrophobic interactions Tyr 178 and Asn 135. Thearubigin was found to interact with RmAChE1 through both hydrogen bonds and hydrophobic interactions. Thearubigin was observed to form hydrogen bonds with Asp 125, Val 123, Ala 208, and Asp 342 amino acids, with bond distances measuring 3.02, 2.66, 3.38, and 2.52 Å.

Tyr 388 and Phe 389 show hydrophobic interaction with thearubigin. Fortunellin exhibited a similar interaction pattern as thearubigin, forming hydrogen bonds with Trp 384, Phe 389, Glu 255, and Gly 175 amino acids, with bond distances of 2.26, 2.57, 1.84, and 2.42 Å. Asp 125, Leu 127, and Tyr 388 show hydrophobic interaction with fortunellin. The results depicted in Figure 5 indicate the interaction mode of A) quercetagetin-7-O-(6-O-caffeoyl-β-D-glucopyranoside), B) isorhamnetin, and C) liquiritin towards the RmTIM protein

Quercetagetin interacts with Lysine 13 and Serine 96 amino acids by forming two hydrogen bonds at distances of 2.77 and 3.49 Å, respectively. In addition, hydrophobic interactions between quercetagetin and the amino acids Thr 172, Lys 174, Ser 211, Gly 233, His 15, and His 100 are also observed. Isorhamnetin forms three hydrogen bonds with Lys 13, Glu 96, and Leu 237 amino acids, with bond distances of 2.79, 2.41, and 2.85 Å, respectively. Additionally, isorhamnetin also shows hydrophobic interactions with Ile 170, Lys 274, Ser 211, and Val 231. Liquiritin is observed to interact with specific amino acid residues in its binding site through both hydrophobic contacts and hydrogen bonds. Specifically, it forms hydrophobic contacts with Lys 174, Ile 170, Glu 239, Ser 235, and Ala 234, as well as five hydrogen bonds with Lys 13, Glu 97, Ser 211, Asn 213, and Lys 237. These interactions occur at distances of approximately 2.81, 2.26, 2.47, 2.32, and 2.65 Å.

Figure 4: . Docking interactions between the RmAChE1 protein and (A) methylenebisphloridzin, (B) thearubigin, and (C) fortunellin

Figure 5: Docking interactions between the RmTIM protein and (A) quercetagetin-7-O-(6-O-caffeoyl-β-D-glucopyranoside) 2, (B) isorhamnetin, and (C) liquiritin

5.Could the authors discuss the stability of these flavonoids in the application?

->Thank you for your kind comment. The stability of these flavonoids has been discussed in the discussion section of the manuscript from the literature.

"The stability and biological activity of flavonoids can be influenced by temperature, and their sensitivity to heat treatment can vary based on their chemical structure. Generally, glycosylated flavonoids exhibit greater resistance to heat treatment compared to aglycone flavonoids [38]. Despite the fact that the mechanism is unknown, the incorporation of acyl groups has been found to increase the thermostability of flavonoids [39]. For instance, Ishihara and Nakajima conducted a study where quercetin-3-glucoside was monoacylated in vitro using nine different aromatic carboxylic acids. The resulting flavonoids showed improved thermostability and light-resistivity [40].

References:

  1. Hind, C. Effect of heat processing on thermal stability and antioxidant activity of six flavonoids. J. Food Process. Preserv. 2017, v. 41, pp. --2017 v.2041 no.2015, doi:10.1111/jfpp.13203.
  2. Plaza, M.; Pozzo, T.; Liu, J.; Gulshan Ara, K.Z.; Turner, C.; Nordberg Karlsson, E. Substituent Effects on in Vitro Antioxidizing Properties, Stability, and Solubility in Flavonoids. J. Agric. Food Chem. 2014, 62, 3321-3333, doi:10.1021/jf405570u.
  3. Ishihara, K.; Nakajima, N. Structural aspects of acylated plant pigments: stabilization of flavonoid glucosides and interpretation of their functions. Journal of Molecular Catalysis B: Enzymatic 2003, 23, 411-417, doi:https://doi.org/10.1016/S1381-1177(03)00106-1.

Reviewer 3 Report

The research is quite weak. The purpose of this research is not entirely clear, because if the goal was to screen flavonoids as potential acaricides for controlling Rhipicephalus microplus then the research is methodologically poorly done. In virtual screening, the protein preparation step is the most crucial; these tests' success usually depends on it. At the outset, a few or a dozen models are considered (obtained from the result of studies of substances with known activity, although an approach such as in this work can eventually be accepted), which are then validated in elimination tests and the model that is best able to distinguish active from inactive substances is selected. In the paper, a crystal structure existed for the TIM protein, which is the most favourable case. Still, even then the effectiveness of such a protein model (the chosen active site) as a potential acaricide target must be confirmed.
I find the manuscript unsuitable for publication.

Author Response

Reviewer 3:

The research is quite weak. The purpose of this research is not entirely clear, because if the goal was to screen flavonoids as potential acaricides for controlling Rhipicephalus microplus then the research is methodologically poorly done. In virtual screening, the protein preparation step is the most crucial; these tests' success usually depends on it. At the outset, a few or a dozen models are considered (obtained from the result of studies of substances with known activity, although an approach such as in this work can eventually be accepted), which are then validated in elimination tests and the model that is best able to distinguish active from inactive substances is selected. In the paper, a crystal structure existed for the TIM protein, which is the most favourable case. Still, even then the effectiveness of such a protein model (the chosen active site) as a potential acaricide target must be confirmed.
I find the manuscript unsuitable for publication.

->Thank you for your professional comments. We have revised the methodology portion and updated it according to your suggestion, as mentioned below:

4.1 Preparation of TIM and Homology modeled of RmAChE1

4.2. Prediction of the active site

4.3. Ligands searching and database preparation

4.4. Docking analysis

4.5. Ligand interaction calculations

Please check the sessions in our revised manuscript. In drug design and discovery, diverse computational chemistry approaches are used to calculate and predict events, such as the drug binding to its target and the chemical properties for designing potential new drugs. So we use a computational approach to find the best flavonoid derivatives against RmAchE1 and TIM protein. Theoretically, this can be utilized as the best drug and will facilitate the scientist in the future.

Reviewer 4 Report

The article uses a number of standard techniques and software tools (such as protein homology modeling and molecular docking) to identify a number of flavonoid-like compounds as potential inhibitors of acetylcholinesterase and triosephosphate isomerase enzymes of Rhipicephalus microplus ticks. Although the general approach is reasonable, each step involves significant uncertainty that undermines the reliability of the predictions. At the same time, the authors provide no experimental validation of the predicted activities. Taking this into account, the article could be published after a MAJOR revision addressing the following issues.

1) The focus on flavonoids as the screening compounds has no justification beyond the assertion that “The interaction of flavonoids with proteins and nucleic acids has resulted in the development of numerous bioactive compounds with pharmacological, antibacterial, and insecticidal properties. Since flavonoids are used as pesticides in medicine and agriculture, they are significant. Therefore, they could be useful in a pest control program”. Moreover, the selection criteria used are not clear (“databases were searched for flavonoid compounds that could operate as TIM and AChE1 inhibitors and are used in insecticides and pesticide”): only 80 compounds were included, and not all of them are in fact flavonoids.

2) It is not clear which templates were used for the homology modeling of the RmAChE1 and RmTIM structures (and in the latter case, to what extent the homology modeling was actually used). Moreover, the location and structure of their active sites (including the key amino acid residues known from other species) was not analyzed and taken into account. For RmAChE1, the automated search apparently was able to detect the active site gorge, but any discussion of the inhibitor binding modes is pretty meaningless when it is not clear which residues constitute the catalytic triad or the peripheral anion site). For RmTIM, the detected pocket is located at the dimer interface, far from the known active sites in each subunit (for this species, they are His95, Glu165, and auxiliary residues Lys13 and the 166-176 loop). Although the dimer interactions in TIMs are indeed known to influence the enzyme stability and activity, this approach to search for inhibitors is not yet validated and can only be seen as secondary to targeting the active site. At any rate, molecular docking is inherently uncertain and cannot “prove their strong interaction”. The authors should provide at least some experimental confirmation of the proposed activities.

3) At least for the cholinesterases, some of the flavonoids and related compounds are indeed known to inhibit the AChE and BChE enzymes from various species including humans and were even considered as potential anti-Alzheimer’s agents. However, their interspecies selectivity is not clear while the concentrations required are rather high (from tens to hundreds μM; see, e.g., https://doi.org/10.1515/znc-2019-0079 and references therein). TIMs also play a crucial role in the metabolism of almost all organisms (from humans and other mammals to bacteria), and ensuring inhibitory selectivity may be difficult due to high inter-species conservation of their key residues. Thus, it is not clear if the proposed flavonoid inhibitors of RmAChE1 and RmTIM could be used as acaricidal agents in an efficient and safe way.

4) It is not clear what is the purpose of the DFT calculations for some of the proposed compounds and the isolated amino acids, and how it contributes to the understanding of the potential protein-ligand interactions. The assertion that “The DFT results also revealed that both compounds have better bioactivity and chemical reactivity with considerable intramolecular charge transfer between electron-donor and electron-acceptor groups” is not justified.

5) English in the article must be substantially improved with respect to misprints, grammar, and style. Some of the assertions are also scientifically questionable. For example, “residual issues”, “tick management techniques, such as natural products and commodities”, “These computationally-driven discoveries are beneficial and can be utilized in assessing drug bioavailability in both in vitro and in vivo settings”, “On the sixth day,… the embryo assumes control over its development”, “secondary metabolites, or flavonoids”, “medicinal chemistry is frequently used to identify new bioactive molecules with acaricidal applications”, “We could use these characteristics to reduce the energy of specific molecules”, “Ramachandran map’s u and w distribution”, “accepter”, “product remains, active component withdrawal”, “plant-based chemicals… have limited human harm, short environmental persistence, and complicated chemistry, making them ideal pesticide candidates”, “Many different pharmacological effects, such as antidiabetic, anti-inflammatory, anti-hyperglycemic, anti-cancer, antibacterial, acetylcholinesterase inhibitory, enzymatic, hepatoprotective, and antioxidant” (no subject or verb), misplaced introductory phrases such as “similarly”, “in contrast”, and “moreover”. The references should more accurately follow the journal style (e.g., the journal names should be abbreviated).

Author Response

Reviewer 4:
The article uses a number of standard techniques and software tools (such as protein homology modeling and molecular docking) to identify a number of flavonoid-like compounds as potential inhibitors of acetylcholinesterase and triosephosphate isomerase enzymes of Rhipicephalus microplus ticks. Although the general approach is reasonable, each step involves significant uncertainty that undermines the reliability of the predictions. At the same time, the authors provide no experimental validation of the predicted activities. Taking this into account, the article could be published after a MAJOR revision addressing the following issues.

->Thank you for your sincere comments. Thank you for the agreement that our general approach is reasonable. We tried our bests to revise our draft point to point according to your comments.

1. The focus on flavonoids as the screening compounds has no justification beyond the assertion that "The interaction of flavonoids with proteins and nucleic acids has resulted in the development of numerous bioactive compounds with pharmacological, antibacterial, and insecticidal properties. Since flavonoids are used as pesticides in medicine and agriculture, they are significant. Therefore, they could be useful in a pest control program". Moreover, the selection criteria used are not clear ("databases were searched for flavonoid compounds that could operate as TIM and AChE1 inhibitors and are used in insecticides and pesticide" ): only 80 compounds were included, and not all of them are in fact flavonoids.

->Thank you for your professional comments. It is true that flavonoids have been shown to interact with proteins and nucleic acids and have been used as pesticides.

2. It is not clear which templates were used for the homology modeling of the RmAChE1 and RmTIM structures (and in the latter case, to what extent the homology modeling was actually used). Moreover, the location and structure of their active sites (including the key amino acid residues known from other species) was not analyzed and taken into account. For RmAChE1, the automated search apparently was able to detect the active site gorge, but any discussion of the inhibitor binding modes is pretty meaningless when it is not clear which residues constitute the catalytic triad or the peripheral anion site). For RmTIM, the detected pocket is located at the dimer interface, far from the known active sites in each subunit (for this species, they are His95, Glu165, and auxiliary residues Lys13 and the 166-176 loop). Although the dimer interactions in TIMs are indeed known to influence the enzyme stability and activity, this approach to search for inhibitors is not yet validated and can only be seen as secondary to targeting the active site. At any rate, molecular docking is inherently uncertain and cannot "prove their strong interaction". The authors should provide at least some experimental confirmation of the proposed activities.

->Thank you for your professional comments. We subsequently revised our manuscript with the following paragraphs.

3.1 Homology Modeled of RmAChE1 and its Validation

The amino acid sequence of RmAChE1, with accession number A0A0F6P2D6, was retrieved from the UniProt database (found at http://www.uniprot.org), which is a comprehensive source for protein sequences. The BLAST analysis revealed the existence of 100 protein sequences that showed significant similarities. Out of these, 48 were Acetylcholinesterases from other species, 21 were Butyrylcholinesterases, and the remaining 31 were proteins that shared some degree of identity with RmAChE1. The crystal structure of mouse acetylcholinesterase (PDB: 5DTI, Chain A) was selected as the template among the 48 AChEs due to its highest resolution of 2 Å. Additionally, this structure showed a high level of similarity to RmAChE1, with 44% identity and 89% query coverage. Validation of the homology model was performed as mentioned in the manuscript.

3.2 Preparation of RmAChE1 and TIM, S Protein.

The same protocol was used as mentioned in the manuscript.

Active Site Prediction

In the context of docking studies, it is imperative to pinpoint the putative binding pockets within the modeled protein structures. The three-dimensional structure of RmAChE1 has not been documented in available databases, resulting in the absence of any reported catalytic sites. As a result, it is imperative to anticipate the potential binding regions of the receptor and determine the area with the largest cavity for the purpose of docking compounds. For this purpose, we utilized the site finder tool integrated into MOE to determine the probable binding sites within RmAChE1. The analysis resulted in the identification of eight high-potential binding pockets. After evaluating the potential binding pockets, we selected the one that comprises the following amino acids: GLN122, VAL123, LEU124, ASP125, THR126, LEU127, SER134, TRP137, ASN138, ALA139, TYR173, GLY174, GLY175, GLY176, TYR178, Ser179, GLY180, THR181, LEU184, TYR187, GLU255, SER256, TRP289, THR335, ASN336, SER337, GLY338, GLY339, VAL340, VAL341, ASP342, PHE343, PRO344, TRP384, and PHE385. In the case of TIM, the catalytic site has been previously reported (Marsh and Shah 2014). As such, we selected the corresponding residues at the catalytic site and performed docking simulations with the related compounds.

3. At least for the cholinesterases, some of the flavonoids and related compounds are indeed known to inhibit the AChE and BChE enzymes from various species including humans and were even considered as potential anti-Alzheimer's agents. However, their interspecies selectivity is not clear while the concentrations required are rather high (from tens to hundreds μM; see, e.g., https://doi.org/10.1515/znc-2019-0079 and references therein). TIMs also play a crucial role in the metabolism of almost all organisms (from humans and other mammals to bacteria), and ensuring inhibitory selectivity may be difficult due to high inter-species conservation of their key residues. Thus, it is not clear if the proposed flavonoid inhibitors of RmAChE1 and RmTIM could be used as acaricidal agents in an efficient and safe way.

->Thank you for your professional comments. This computational tool assesses the "drug-likeness" of a molecule that plays a vital role in helping chemists effectively screen drug targets (RmAChE1 and RmTIM) before validating it experimentally. 

4. It is not clear what is the purpose of the DFT calculations for some of the proposed compounds and the isolated amino acids, and how it contributes to the understanding of the potential protein-ligand interactions. The assertion that "The DFT results also revealed that both compounds have better bioactivity and chemical reactivity with considerable intramolecular charge transfer between electron-donor and electron-acceptor groups" is not justified.

->Thank you for your professional comments. Based on the computational approach to determine the chemical reactivity of both compounds, it is mandatory to confirm the chemical bonding type and parameters necessary to achieve legitimate properties of compounds. This study includes qualitative computational analysis about forming a bond between amino acids of RmACHE1 and Liquiritin & Isorhamnetin, respectively. Bond lengths and MEP map confirmed hydrogen bonding between complex forming molecules. This information is sufficient to conclude that formed complexes are chosen after satisfying results. The bioactivity of all complexes was analyzed by docking, and chemical properties supported and provided further information about the chemical makeup of molecules being used with the help of a net charge contour map which showed that molecules were stable in selected positions. Charge transfer between these molecules through Hydrogen bonding gave the values of bond lengths between bonding atoms, and bonding between electron-donor and electron-acceptor groups was confirmed by resultant values. Overall, DFT calculations played a special role in exploring additional information about the chemical mechanism followed in bioactive compounds' activities.

5. English in the article must be substantially improved with respect to misprints, grammar, and style. Some of the assertions are also scientifically questionable. For example, "residual issues", "tick management techniques, such as natural products and commodities", "These computationally-driven discoveries are beneficial and can be utilized in assessing drug bioavailability in both in vitro and in vivo settings", "On the sixth day,… the embryo assumes control over its development", "secondary metabolites, or flavonoids", "medicinal chemistry is frequently used to identify new bioactive molecules with acaricidal applications", "We could use these characteristics to reduce the energy of specific molecules", "Ramachandran map's u and w distribution", "accepter", "product remains, active component withdrawal", "plant-based chemicals… have limited human harm, short environmental persistence, and complicated chemistry, making them ideal pesticide candidates", "Many different pharmacological effects, such as antidiabetic, anti-inflammatory, anti-hyperglycemic, anti-cancer, antibacterial, acetylcholinesterase inhibitory, enzymatic, hepatoprotective, and antioxidant" (no subject or verb), misplaced introductory phrases such as "similarly", "in contrast", and "moreover". The references should more accurately follow the journal style (e.g., the journal names should be abbreviated).Line 45: It should be "druggable ATX-LPAR axis"?

->Thank you for your professional comments. We tried our best regarding the critique of the English language in this manuscript. The references are formatted in accordance with the journal's guidelines, including the correct abbreviation of journal titles.

Round 2

Reviewer 2 Report

The authors have answered all my questions. I think it's ready to be accepted.

Author Response

Dear professor, thank your for your kindness and sincere comments.

Reviewer 3 Report

The authors have significantly improved the manuscript. Thank you very much, as it is now easier to follow the article.
However, I still have one comment regarding the fragment:
"The 3D structure of RmAChE1 has not been documented in available databases, resulting in the absence of any reported catalytic sites. As a result, it is imperative to anticipate the potential binding regions of the receptor and determine the area with the largest cavity for docking compounds. For this purpose, we utilised the site finder tool integrated into MOE to determine the probable binding sites within RmAChE1. The analysis resulted in the identification of eight high-potential binding pockets. After evaluating the potential binding pockets, we selected the one that comprises the following amino acids:..."
I would ask the authors to verify the binding sites in a test of activity predictability, i.e. distinguishing active from inactive compounds by the assumed model. Recommended literature: 10.1186/s13321-015-0100-8
I consider that when the mechanism of inhibitor binding at the molecular level is not known, the approach proposed by the authors is insufficient.

Author Response

Reviewer 3:

  1. The authors have significantly improved the manuscript. Thank you very much, as it is now easier to follow the article. However, I still have one comment regarding the fragment: "The 3D structure of RmAChE1 has not been documented in available databases, resulting in the absence of any reported catalytic sites. As a result, it is imperative to anticipate the potential binding regions of the receptor and determine the area with the largest cavity for docking compounds. For this purpose, we utilised the site finder tool integrated into MOE to determine the probable binding sites within RmAChE1. The analysis resulted in the identification of eight high-potential binding pockets. After evaluating the potential binding pockets, we selected the one that comprises the following amino acids:..."

I would ask the authors to verify the binding sites in a test of activity predictability, i.e. distinguishing active from inactive compounds by the assumed model. Recommended literature: 10.1186/s13321-015-0100-8

I consider that when the mechanism of inhibitor binding at the molecular level is not known, the approach proposed by the authors is insufficient.

-> We are very grateful for your professional comments and advice for a better manuscript. Per your professional advice, we revised the relevant part of our manuscript. Moreover, the amendments are highlighted in red in the revised manuscript. As we have already mentioned, the model protein's active site was determined by the Site-Finder tool mentioned in the manuscript. Moreover, it was further verified by reviewer 3’s suggestion using the protocol mentioned in the recommended literature (10.1186/s13321-015-0100-8). Active compounds were docked in the mentioned active sites. So, all active sites were perfectly docked into the proposed active site, as shown in Figure 3 (below). Additionally, inactive could not be docked in the proposed active site.

Figure 3: Active compounds were docked in the mentioned active sites.

Reviewer 4 Report

The authors have significantly extended and improved the article. However, some of the reviewer’s comments have not yet adequately addressed.

1. The focus on flavonoids as the screening compounds has no justification beyond the assertion that "The interaction of flavonoids with proteins and nucleic acids has resulted in the development of numerous bioactive compounds with pharmacological, antibacterial, and insecticidal properties. Since flavonoids are used as pesticides in medicine and agriculture, they are significant. Therefore, they could be useful in a pest control program". Moreover, the selection criteria used are not clear ("databases were searched for flavonoid compounds that could operate as TIM and AChE1 inhibitors and are used in insecticides and pesticide"): only 80 compounds were included, and not all of them are in fact flavonoids.

->Thank you for your professional comments. It is true that flavonoids have been shown to interact with proteins and nucleic acids and have been used as pesticides.

While true, this does not explain why flavonoids should be expected to inhibit specifically TIM and AChE, or why only 80 compounds were included while not all of them are in fact flavonoids.

2. Regarding the homology modeling and active site detection. For RmAChE1, the relevant information has been included; however, a general view of the active site gorge and the positions of the key residues should also be clearly shown. It is not reasonable to require a reader to look up or decode the numbering of the catalytic triad for this enzyme (Ser256, Glu381, His494) and the bound ligand positions in the gorge. For RmTIM, the binding site actually used for docking and shown in Fig. 5 is correct; however, an entirely different pocket at the dimer interface is marked in Fig. 3. Obviously, this must be corrected! Moreover, the protein X-ray structure (PDB: 3TH6) apparently was used as-is (except water removal), even the missing loop between Ile170 and Ala176 was not reconstructed, but Fig. 1 refers to “the structural models predicted using the online SWISS-MODEL tool”.

The authors should also provide at least some experimental confirmation of the proposed activities as the molecular docking results are inherently uncertain.

In the Fig. S2-S5, it is not correct to speak of “molecular docking of RmAChE1” (a ligand is normally docked into a protein) or the “reacting sites” (no chemical reaction takes place).

3. At least for the cholinesterases, some of the flavonoids and related compounds are indeed known to inhibit the AChE and BChE enzymes from various species including humans and were even considered as potential anti-Alzheimer's agents. However, their interspecies selectivity is not clear while the concentrations required are rather high (from tens to hundreds μM; see, e.g., https://doi.org/10.1515/znc-2019-0079 and references therein). TIMs also play a crucial role in the metabolism of almost all organisms (from humans and other mammals to bacteria), and ensuring inhibitory selectivity may be difficult due to high inter-species conservation of their key residues. Thus, it is not clear if the proposed flavonoid inhibitors of RmAChE1 and RmTIM could be used as acaricidal agents in an efficient and safe way.

->Thank you for your professional comments. This computational tool assesses the "drug-likeness" of a molecule that plays a vital role in helping chemists effectively screen drug targets (RmAChE1 and RmTIM) before validating it experimentally.

It is not clear what tool is mentioned here and how this addresses the issues of the selectivity and safety of the proposed inhibitors.

4. The goal and relevance of the DFT analysis still have not been justified. The model amino acids should at least be found in the binding sites and their orientation and position should at least be consistent. Moreover, some of the hydrogen bonds shown in Fig. 6-7 are impossible as the carboxylate hydroxyl groups are not present in the protein (and in a free amino acid they are fully ionized). It is not correct to use free neutral amino acids to model amino acid residues within a protein! Alternatively, this section could be removed as it does not really improve the understanding of ligand interactions.

Author Response

Reviewer 4:

The authors have significantly extended and improved the article. However, some of the reviewer’s comments have not yet adequately addressed.

-> Thank you for your professional comments, we sincerely appreciate your expertise.

  1. The focus on flavonoids as the screening compounds has no justification beyond the assertion that "The interaction of flavonoids with proteins and nucleic acids has resulted in the development of numerous bioactive compounds with pharmacological, antibacterial, and insecticidal properties. Since flavonoids are used as pesticides in medicine and agriculture, they are significant. Therefore, they could be useful in a pest control program". Moreover, the selection criteria used are not clear ("databases were searched for flavonoid compounds that could operate as TIM and AChE1 inhibitors and are used in insecticides and pesticide"): only 80 compounds were included, and not all of them are in fact flavonoids.

While true, this does not explain why flavonoids should be expected to inhibit specifically TIM and AChE, or why only 80 compounds were included while not all of them are in fact flavonoids.

-->  We appreciate the reviewers' interest in selecting the flavonoids used in our study. It is true that flavonoids have been shown to interact with proteins and nucleic acids and have been used as pesticides. The selection of these particular flavonoids was based on a variety of factors. Firstly, we conducted an extensive literature review to identify flavonoids that have been previously reported to exhibit potential bioactivity against different parasites and insects. We then selected flavonoids that had strong evidence of bioactivity in previous studies and that possessed chemical structures that were readily available in the PubChem database. Brief details are mentioned under the session “4.3. Ligands searching and database preparation”.

In addition to bioactivity and chemical structure, we also considered the availability of flavonoids. We selected readily available flavonoids.

Finally, we used computational methods to refine our selection of flavonoids further. We conducted molecular docking analyses to predict the potential binding of the flavonoids to the acetylcholinesterase and triose-phosphate isomerase proteins of the Rhipicephalus microplus. Moreover, based on their docking scores, 9 key flavonoids were discussed in the study. The comment has been addressed and incorporated into the manuscript. Thank you for your advice.

  1. Regarding the homology modeling and active site detection. For RmAChE1, the relevant information has been included; however, a general view of the active site gorge and the positions of the key residues should also be clearly shown. It is not reasonable to require a reader to look up or decode the numbering of the catalytic triad for this enzyme (Ser256, Glu381, His494) and the bound ligand positions in the gorge. For RmTIM, the binding site actually used for docking and shown in Fig. 5(Fig. 6) is correct; however, an entirely different pocket at the dimer interface is marked in Fig. 3(Fig 4). Obviously, this must be corrected! Moreover, the protein X-ray structure (PDB: 3TH6) apparently was used as-is (except water removal), even the missing loop between Ile170 and Ala176 was not reconstructed, but Fig. 1 refers to “the structural models predicted using the online SWISS-MODEL tool”.

The authors should also provide at least some experimental confirmation of the proposed activities as the molecular docking results are inherently uncertain.

In the Fig. S2-S5, it is not correct to speak of “molecular docking of RmAChE1” (a ligand is normally docked into a protein) or the “reacting sites” (no chemical reaction takes place).

--> We replaced the Fig. 4 entitled “Figure 4. (A) RmAChE1 and (B) RmTIM’s 3D structures showed active sites by small red and grey dots” with a new one that exhibited the binding pocket present in chain A of the RmTIM. Some residues were missing in the structure of TIM,s so we constructed the missing residues by using a loop modeler implemented in MOE software. TIM's 3D structure [PDB: 3TH6] was retrieved from the PDB archive of the Research Collaboratory for Structural Bioinformatics (RCSB) (https://www.rcsb.org). The 3D structure of the RmAChE1 protein was not reported. Therefore, we needed to model the structure of RmAChE1. The amino acid sequence of RmAChE1 was retrieved from uniport https://www.uniprot.org/uniprotkb/ under accession No. A0A0F6P2D6 RHIMP and then modeled using SWISS-MODEL https://swissmodel.expasy.org/interactive/.

B

Figure 4. (A) RmAChE1 and (B) RmTIM’s 3D structures showed active sites by small red and grey dots.

--> Yes, we agree with your addressing points based on some experimental confirmation of the proposed activities. However, a in vitro study is another ongoing project to assess the toxicological studies of flavonoids against RmAChE1 or against RmTIM to verify the accuracy of these predicted results better. We will conduct experiments to verify their bioactivities and publish the results soon.

--> We corrected Fig. S2-S5 according to reviewer 4’s instruction as mentioned under the heading of supplementary materials in the manuscript.

  1. At least for the cholinesterases, some of the flavonoids and related compounds are indeed known to inhibit the AChE and BChE enzymes from various species including humans and were even considered as potential anti-Alzheimer's agents. However, their interspecies selectivity is not clear while the concentrations required are rather high (from tens to hundreds μM; see, e.g., https://doi.org/10.1515/znc-2019-0079 and references therein). TIMs also play a crucial role in the metabolism of almost all organisms (from humans and other mammals to bacteria), and ensuring inhibitory selectivity may be difficult due to high inter-species conservation of their key residues. Thus, it is not clear if the proposed flavonoid inhibitors of RmAChE1 and RmTIM could be used as acaricidal agents in an efficient and safe way.

--> Thank you for your professional comments. The computational tool assesses the "drug-likeness" of a molecule that plays a vital role in helping chemists effectively screen drug targets (RmAChE1 and RmTIM) before validating it experimentally. Drug-like seems to be a potential paradigm to codify the balance between a compound's molecular characteristics affecting its pharmacodynamics and pharmacokinetics and ultimately optimize its absorption, distribution, metabolism, and excretion (ADME) in the human body like a drug. We incorporated the drug-like calculation based on the Lipinski rule of five to determine oral absorption or membrane permeability. We used the new SwissADME web tool, available at http://www.swissadme.ch, which provides free access to collect accurate predictive models for drug-likeness, pharmacokinetics, physicochemical features, and medicinal chemistry friendliness. The physicochemical properties of flavonoid molecules are presented in Table 2.

  1. The goal and relevance of the DFT analysis still have not been justified. The model amino acids should at least be found in the binding sites and their orientation and position should at least be consistent. Moreover, some of the hydrogen bonds shown in Fig. 6-7 are impossible as the carboxylate hydroxyl groups are not present in the protein (and in a free amino acid they are fully ionized). It is not correct to use free neutral amino acids to model amino acid residues within a protein! Alternatively, this section could be removed as it does not really improve the understanding of ligand interactions.

--> We are very grateful for your professional comments and advice for a better manuscript. DFT results revealed that both compounds (isorhamnetin and liquiritin) have better chemical reactivity and kinetic stability with considerable intramolecular charge transfer between electron-donor and electron-acceptor groups. The molecule with a higher band gap was less polarizable and showed high kinetic stability and low chemical reactivity (hard molecule). The band gap shown by the complex was lower than the obtained band gaps of liquiritin, making it more reactive towards the affected area. While in the case of isorhamnetin, the band gap of complexes was more than isorhamnetin itself, making it stable reactive. This much smaller band gap of complex predicts the need for a small quantity of energy to get excited from ground level.
